# Extrafollicular CD4+ T-B interactions are sufficient for inducing autoimmune-like chronic graft-versus-host disease

Ruishu Deng[1,2,8], Christian Hurtz[3,9], Qingxiao Song[1,2,4], Chanyu Yue[5], Gang Xiao[3,10], Hua Yu[5], Xiwei Wu[6], Markus Muschen[3,10], Stephen Forman[2], Paul J. Martin[7] & Defu Zeng [1,2]

Chronic graft-versus-host disease (cGVHD) is an autoimmune-like syndrome mediated by pathogenic CD4+ T and B cells, but the function of extrafollicular and germinal center CD4+ T and B interactions in cGVHD pathogenesis remains largely unknown. Here we show that extrafollicular CD4+ T and B interactions are sufficient for inducing cGVHD, while germinal center formation is dispensable. The pathogenesis of cGVHD is associated with the expansion of extrafollicular CD44$^{hi}$CD62$^{lo}$PSGL-1$^{lo}$CD4+ (PSGL-1$^{lo}$CD4+) T cells. These cells express high levels of ICOS, and the blockade of ICOS/ICOSL interaction prevents their expansion and ameliorates cGVHD. Expansion of PSGL-1$^{lo}$CD4+ T cells is also prevented by BCL6 or Stat3 deficiency in donor CD4+ T cells, with the induction of cGVHD ameliorated by BCL6 deficiency and completely suppressed by Stat3 deficiency in donor CD4+ T cells. These results support that Stat3- and BCL6-dependent extrafollicular CD4+ T and B interactions play critical functions in the pathogenesis of cGVHD.

[1] Diabetes and Metabolism Research Institute, The Beckman Research Institute of City of Hope, Duarte, CA 91010, USA. [2] Hematologic Malignancies and Stem Cell Transplantation Institute, The Beckman Research Institute of City of Hope, Duarte, CA 91010, USA. [3] Department of Laboratory Medicine, University of California, San Francisco, CA 94143, USA. [4] Department of Hematology, Fujian Institute of Hematology, Fujian Medical University Union Hospital, Fuzhou 350000, China. [5] Department of Cancer Immunotherapeutic and Tumor Immunology, The Beckman Research Institute of City of Hope, Duarte, CA 91010, USA. [6] Department of Molecular and Cellular Biology, The Beckman Research Institute of City of Hope, Duarte, CA 91010, USA. [7] Fred Hutchinson Cancer Research Center, University of Washington, Seattle, WA 98109, USA. [8] Present address: Sanford Burnham Prebys Medical, Discovery Institute, La Jolla, CA 92307, USA. [9] Present address: Department of Medicine, Division of Hematology and Oncology, University of Pennsylvania, Philadelphia, PA 19104, USA. [10] Present address: Department of Systems Biology, The Beckman Research Institute of City of Hope, Duarte, CA 91010, USA. Correspondence and requests for materials should be addressed to D.Z. (email: dzeng@coh.org)

Allogeneic hematopoietic cell transplantation (HCT) is a curative therapy for hematological malignancies, certain hereditary disorders, and refractory autoimmune diseases[1]. Chronic graft-versus-host disease (cGVHD) remains a major obstacle to the success of this treatment[2, 3]. Chronic GVHD presents with multi-organ pathology and common diagnostic features, as outlined by the NIH consensus criteria. Manifestations include skin pathology varying from lichen planus-like lesions to extensive cutaneous sclerosis, bronchiolitis obliterans as well as salivary and lacrimal gland pathology[4]. Chronic GVHD is an autoimmune-like syndrome caused by the interactions of donor CD4[+] T and B cells and production of IgG[2, 5–9]. Chronic GVHD often follows acute GVHD. The pathogenic autoreactive CD4[+] T cells in cGVHD can derive from CD4[+] T cells in the graft or from T cells generated de novo in a thymic environment damaged by acute GVHD[7]. Due to the destructive effect of alloreactive and autoreactive T cells and IgG antibodies, cGVHD recipients often have lymphopenia at the disease onset[9–11]. This feature differs from other autoimmune diseases (for example, systemic lupus, multiple sclerosis, and type 1 diabetes) that usually have increased numbers of lymphocytes in lymphoid tissues at disease onset[12].

IgG antibody production by B cells requires CD4[+] T-cell help[13]. CD4[+] T- and B-cell interactions occur as multistage and multifactorial processes at the extrafollicular T–B border and in follicular germinal centers (GC)[14]. GC formation requires T- and B-cell expression of BCL6[15]. In brief, naive CD4[+] T cells interact with dendritic cells (DC) in the T-cell zone of a lymphoid follicle and differentiate into Th1, Th2, Th17, and pre-Tfh under different cytokine and microenvironment regulation. Under the influence of IL-6 and ICOS signaling, CD4[+] T cells upregulate the expression of Stat3 and BCL6, and subsequently upregulate the expression of CXCR4, CXCR5, and IL-21, downregulate the expression of CCR7 and PSGL-1(P-selectin glycoprotein ligand 1), and differentiate into pre-Tfh[14]. CCR7 (a ligand for CCL19 and CCL21) and PSGL-1 help anchor T cells to CCL19 and CCL21[16]. Downregulation of CCR7 and PSGL-1 allows the pre-Tfh cells to migrate out of the T-cell zone and reach the T–B border to interact with B cells. This first stage of T–B interaction leads to the generation of short-lived plasma cells and production of low-affinity IgG1, and results in Immunoglobulin Isotype switching without somatic hypermutation[17–19]. In response to CXCL13 (a CXCR5 ligand) from follicular DCs, the CXCR5[hi] pre-Tfh cells migrate further into the center of the B-cell zone to form GCs[20, 21], where the Tfh and B-cell interaction results in somatic hypermutation, production of high affinity IgG, and formation of long-lived plasma cells[20, 22].

Extrafollicular and follicular GC CD4[+] T- and B-cell interactions have an important function in immune defense against infections[14, 20, 23]. Aberrant extrafollicular and follicular T–B interactions have been observed in autoimmune diseases[20, 24, 25]. For example, increased frequencies of Tfh or Tfh-like cells (CXCR5[+]PD-1[hi] or ICOS[hi]) are observed in the spleen of systemic autoimmune Roquin[san/san] mice[24] and in the blood of certain patients with autoimmune Sjogren's syndrome[26]. Mice with systemic lupus have reduced numbers of Tfh in the spleen, but the numbers of extrafollicular PSGL-1[lo]CXCR4[hi]CD4[+] T cells are increased[25]. In keeping with these observations, ectopic clusters of Tfh-like cells and B cells have been identified in the inflamed kidney tissues of patients with systemic lupus erythematosus[27].

Enlargement of GCs and expansion of Tfh and GC B cells have been noted in the spleens of cGVHD mice in different donor → recipient strain combinations, including C57BL/6 (H-2[b]) → B10.BR (H-2[K]), LP/J (H-2[bc]) → B6 and B10.D2 (H-2[d]) → BALB/c (H-2[d]). Results from these studies indicate that GC formation is required for the persistence of cGVHD, suggesting that, like certain autoimmune diseases, the aberrant expansion of Tfh and B cells has an important function in cGVHD pathogenesis[28–30]. On the other hand, patients with active cGVHD have decreased numbers of Tfh-like cells in the blood[31], and Tfh-like cells from the blood of cGVHD patients have an enhanced ability to stimulate IgG antibody production in vitro[32]. GC formation is usually associated with immunoglobulin somatic hypermutation[14], but previous studies showed that allogeneic HCT recipients have gradual recovery of serum IgM, IgG1, and IgG3, but not IgG2 or IgA, and have reduced—immunoglobulin somatic hypermutation at 1 year after HCT, suggesting a lack of GC formation[33, 34]. Moreover, cGVHD onset is associated with lymphopenia in animal models and patients[9–11, 35]. Finally, in a cGVHD model with DBA/2 (H-2[d]) donors and MHC-matched BALB/c recipients, we showed that although IgG antibody-producing B cells and their IgG antibodies have an important function in perpetuating skin damage in cGVHD recipients, lymphoid follicles are destroyed, and GCs are not observed[9]. Taken together, these conflicting results indicate that the function of GC formation in cGVHD pathogenesis remains unclear.

After HCT with grafts containing low numbers of C57BL/6 donor splenocytes, BALB/c recipients survive acute GVHD and then develop chronic GVHD characterized by scleroderma, lymphocytic bronchitis, and damage in the salivary and lacrimal glands, which reflect features of human chronic GVHD[7, 36]. In the current study, we used this model to explore the function of follicular and extrafollicular CD4[+] T and B interactions in the pathogenesis of cGVHD. Our results indicate that extrafollicular T- and B-cell interactions have a critical function, whereas GCs are dispensable in the pathogenesis of cGVHD.

## Results

**Onset of cGVHD and lymphopenia is associated with the absence of GCs.** To address the role of GC formation in cGVHD pathogenesis, BALB/c recipients were injected with T-cell-depleted bone marrow (TCD-BM) plus $1 \times 10^6$ or $0.01 \times 10^6$ spleen cells from C57BL/6 donors. Recipients given TCD-BM alone showed no signs of GVHD (no-cGVHD). Recipients given $1 \times 10^6$ spleen cells developed severe clinical cGVHD with hair loss, and only ~40% survived for more than 60 days (severe cGVHD) (Fig. 1a–c). Recipients given $0.01 \times 10^6$ spleen cells showed mild signs of cGVHD (Fig. 1a–c) and mild reduction of thymic CD4[+]CD8[+] thymocytes (Supplementary Fig. 1).

Lymphoid follicles and GCs in the spleen of the recipients were measured with immunofluorescent staining of tissue sections. We used anti-B220 to visualize B-cell zones, anti-CD3 to visualize T-cell zones and peanut agglutinin (PNA) to identify GCs (Fig. 1d). The GC areas and numbers were measured at days 15, 30, and 60 after HCT. We found no GCs at 15 days after HCT in any recipients (Supplementary Fig. 2). Well-formed GCs were present in no-GVHD recipients at 30 and 60 days after HCT (Supplementary Fig. 3 and Fig. 1d). Recipients with mild cGVHD also had GC formation, slightly reduced in size and numbers, as compared to no-GVHD recipients (Fig. 1d, e and Supplementary Fig. 3). GC enlargement was not observed in any recipients at any time. In contrast, no GCs were observed at any time in recipients with severe cGVHD (Fig. 1d, Supplementary Figs 2 and 3).

To further validate these initial observations, additional experiments were carried out in three other donor–recipient strain combinations[28–30]. In the MHC-matched LP/J → C57BL/6 model, recipients were divided into two groups, severe cGVHD and mild cGVHD, based on the severity of cutaneous cGVHD, survival, and histopathology (Supplementary Fig. 4A–D). We measured GC formation in the spleen at 60 days after HCT.

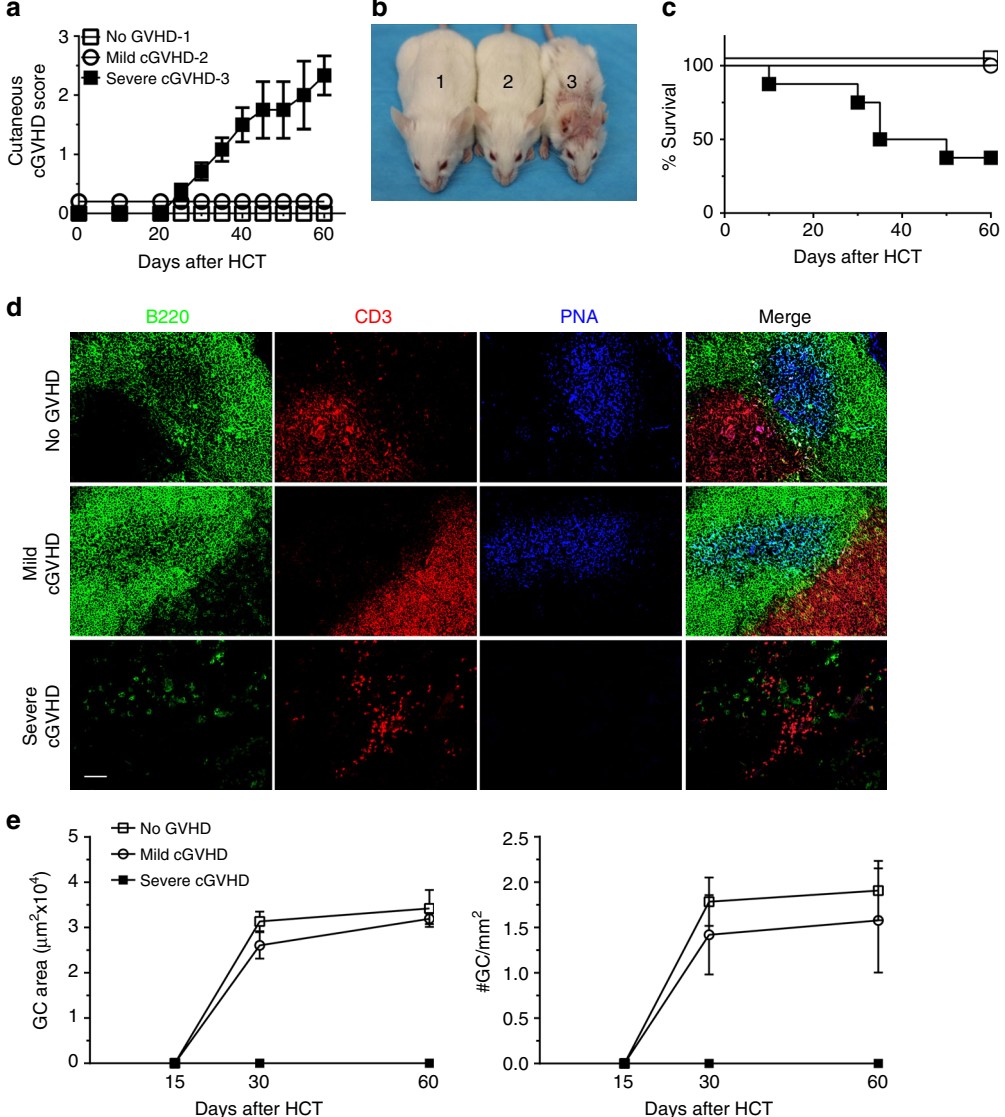

**Fig. 1** No germinal center formation is observed in cGVHD recipients. BALB/c mice were irradiated (850 cGy) and given $2.5 \times 10^6$ TCD-BM alone ($n = 6$) or $2.5 \times 10^6$ TCD-BM plus $1 \times 10^6$ ($n = 8$) or $0.01 \times 10^6$ ($n = 8$) splenocytes from C57BL/6 donors. Mice were monitored for cGVHD. **a** Cutaneous cGVHD score (severe cGVHD group versus no-GVHD group: $P < 0.001$, severe cGVHD group versus mild cGVHD group: $P < 0.001$, two-way ANOVA). **b** Picture taken on day 60 after transplantation (1–no GVHD, 2–mild cGVHD, 3–severe cGVHD). **c** Survival curve (severe cGVHD group versus no GVHD group: $P < 0.001$, severe cGVHD group versus mild cGVHD group: $P < 0.001$, log-rank test). **d** Immunofluorescent staining of B220, CD3, and PNA on cryosections of spleen harvested on day 60 after HCT. **e** Germinal center number and area were measured and are shown as mean ± SE ($n = 6$). *Scale bar,* 50 μm

Compared to no-GVHD recipients given TCD-BM alone, recipients with severe cGVHD showed destruction of lymphoid follicles and absence of GCs (Supplementary Fig. 4E). Recipients with mild cGVHD had lymphoid follicles and GC formation, but the GCs appeared to be smaller than in no-GVHD recipients (Supplementary Fig. 4E). These results were consistent with flow cytometry showing decreased percentages of CXCR5$^{hi}$PD-1$^{hi}$ T$_{FH}$ and Fas$^+$GL7$^+$ GC B cells in splenocytes from recipients with cGVHD (Supplementary Fig. 4F, G). Results of immunofluorescent tissue staining were similar in experiments with the C57BL/6 → B10.BR model (Supplementary Fig. 5A).

To further validate our staining of GCs, we used the combination of IgM and PNA or IgD and GL7 to visualize GCs, as used by others[30, 37]. In the C57B/6 → B10.BR model, smaller GCs were observed in recipients with mild cGVHD, and no GCs were observed in the recipients with severe cGVHD

(Supplementary Fig. 5B). Additionally, in LP/J → C57BL/6 model, no-GVHD recipients given TCD-BM alone showed lymphoid follicles and GCs, but cGVHD recipients showed destruction of lymphoid follicles and absence of GCs under two different staining combinations (Supplementary Fig. 6A, B). Chronic GVHD was also associated with destruction of lymphoid follicles and absence of GCs in the MHC-matched B10.D2 → BALB/c model[38] (Supplementary Fig. 7). Taken together, these results show that cGVHD is associated with destruction of lymphoid follicles and absence of GCs.

**Chronic GVHD is induced in the absence of GC formation.** We tested whether induction of cGVHD requires GC formation. Mice with BCL6 deficiency in T or B cells cannot form GCs[15, 39]. BCL6$^{fl/fl}$Mb1-Cre$^{+/-}$C57BL/6 mice have BCL6 deficiency specifically in B cells (B-BCL6$^{-/-}$)[40]. TCD-BM and spleen cells from B-

BCL6[−/−] or B-BCL6[+/+] control littermates in the C57BL/6 background were transplanted into lethally irradiated BALB/c recipients as described in "Methods" section and previous publication[7]. Recipients given B-BCL6[+/+] or B-BCL6[−/−] TCD-BM cells ($2.5 \times 10^6$) alone showed no signs of GVHD (B-BCL6[+/+] or B-BCL6[−/−] no GVHD). Recipients given additional spleen cells ($1 \times 10^6$) from B-BCL6[+/+] or B-BCL6[−/−] mice developed severe cutaneous cGVHD with hair loss (B-BCL6[+/+]– or

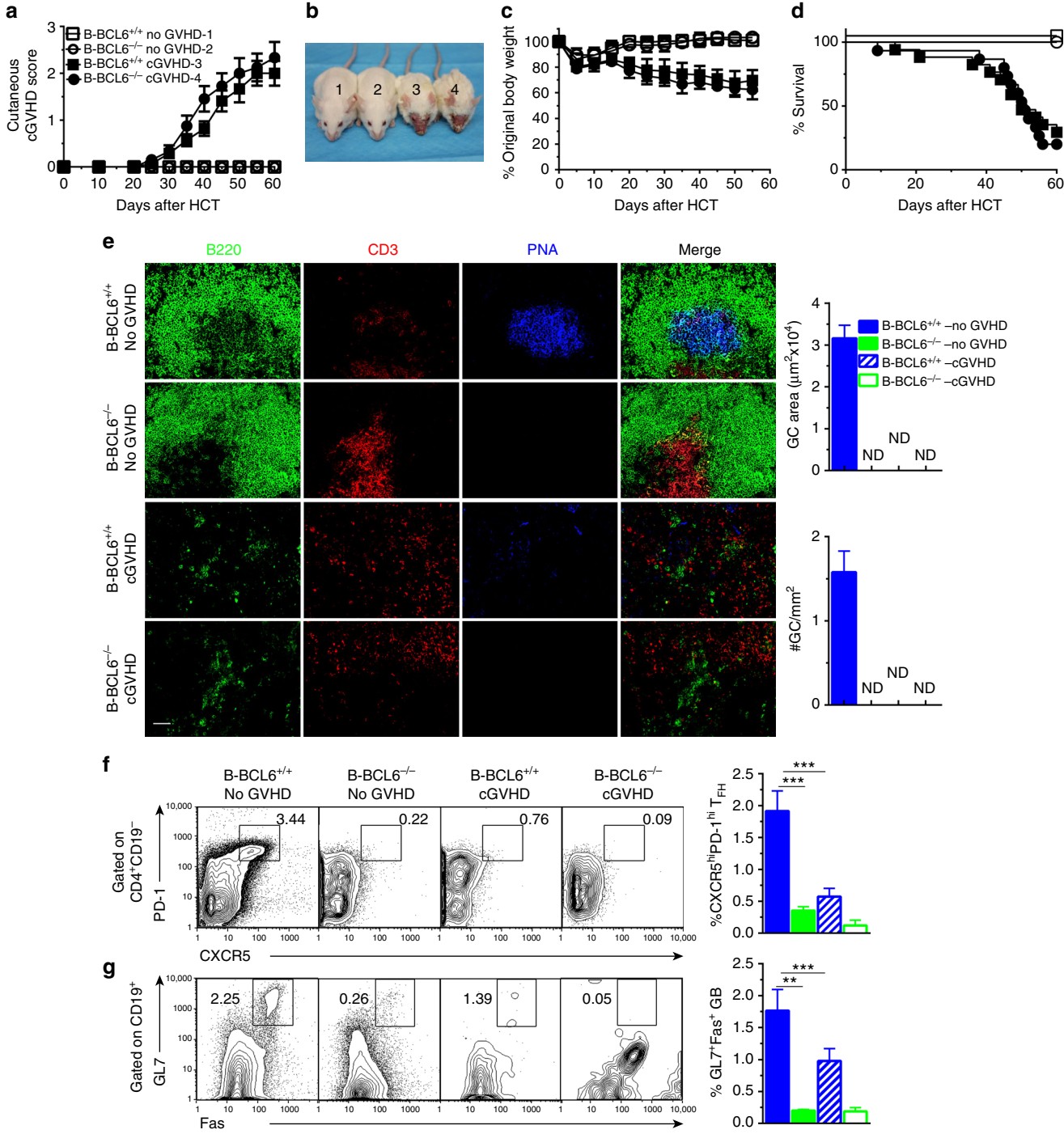

**Fig. 2** Chronic GVHD was induced in recipients without germinal center formation. BALB/c recipients were irradiated (850 cGy) and given $2.5 \times 10^6$ TCD-BM alone ($n = 12$) or $2.5 \times 10^6$ TCD-BM plus $1 \times 10^6$ ($n = 12$) splenocytes from either WT C57BL/6 or BCL6[fl/fl] Mb1-Cre C57BL/6 donors. cGVHD development was monitored. **a** Cutaneous cGVHD score. **b** Picture taken at day 60 after HCT (1 and 3–B-BCL6[+/+] no GVHD and cGVHD, 2 and 4–B-BCL6[−/−] no GVHD and cGVHD). **c** Percent body weight changes. **d** Survival curve. **e** Sixty days after transplantation, spleens were harvested and germinal centers were identified by immunofluorescent staining of B220, CD3, and PNA, and GC area and numbers were measured and are shown as mean ± SE ($n = 6$). **f** Donor splenocytes were stained for CD4, CD19, CXCR5, and PD-1. Tfh were gated as CD4[+]CD19[−] and are shown as CXCR5[hi]PD-1[hi]. Percentages of CXCR5[hi]PD-1[hi] cells among CD4[+]CD19[−] cells were shown as mean ± SE ($n = 6$). **g** Donor splenocytes were stained for CD19, GL7, and Fas. Germinal center B cells were gated on CD19[+] and are shown as GL7[+]Fas[+]. Percentages of GL7[+]Fas[+] among CD19[+] cells are shown as mean ± SE ($n = 6$). **$P < 0.01$, ***$P < 0.001$, unpaired two-tailed Student's $t$ test. B-BCL6[+/+] no GVHD = B-BCL6[+/+] TCD-BM; B-BCL6[−/−] no GVHD = B-BCL6[−/−] TCD-BM; B-BCL6[+/+] cGVHD = B-BCL6[+/+] TCD-BM + B-BCL6[+/+] splenocytes; B-BCL6[−/−] cGVHD = B-BCL6[−/−] TCD-BM + B-BCL6[−/−] splenocytes. *Scale bar*, 50 μm

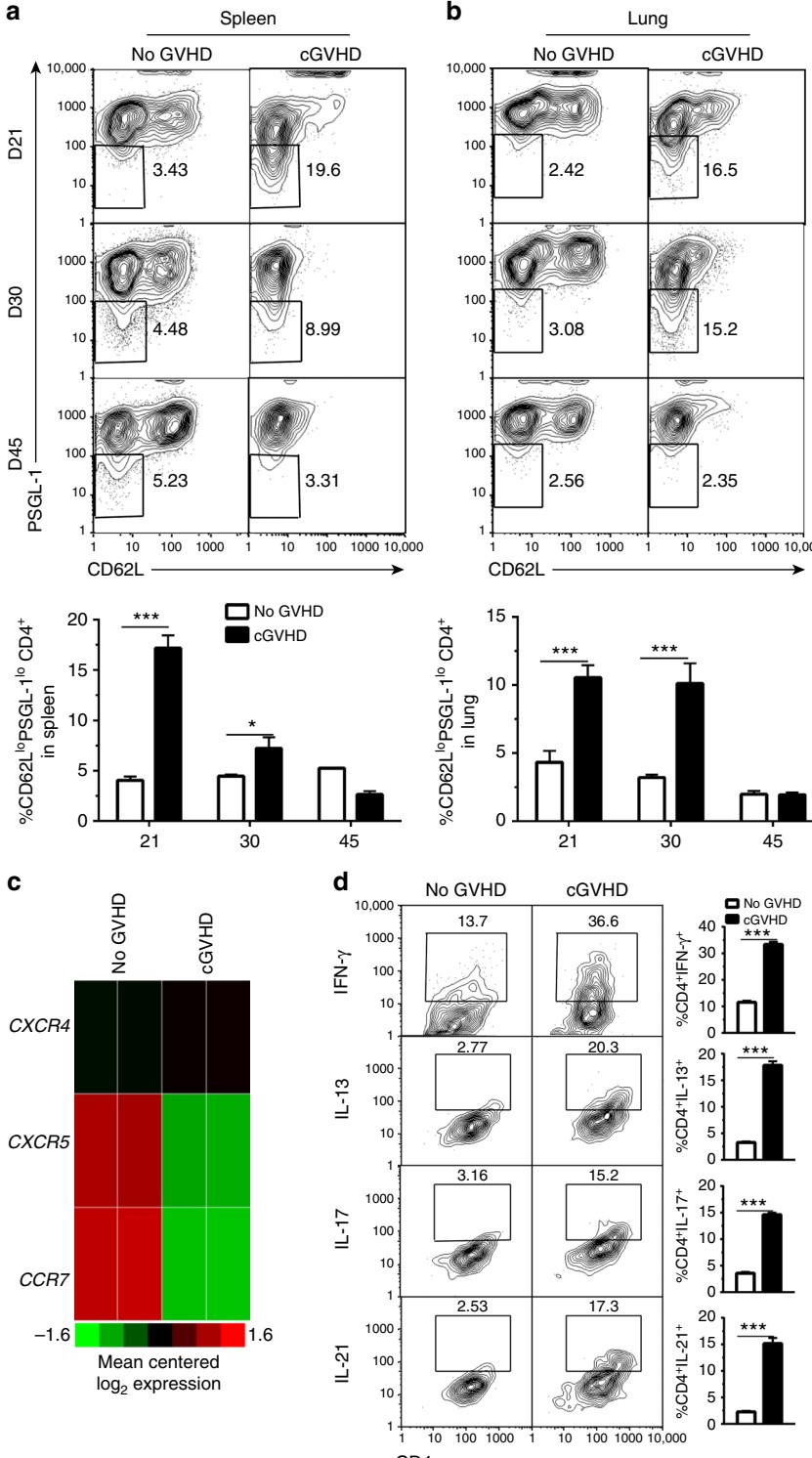

**Fig. 3** cGVHD is associated with expansion of PSGL-1$^{lo}$CD4$^+$ T cells. BALB/c recipients were irradiated (850 cGy) and given $2.5 \times 10^6$ TCD-BM or $2.5 \times 10^6$ TCD-BM plus $1 \times 10^6$ splenocytes from C57BL/6 donors. **a**, **b** Twenty-one, 30, and 45 days after HCT, spleen and lung were harvested. Splenocytes and mononuclear cells isolated from lung were stained for CD4, CD44, PSGL-1, and CD62L. Gated CD4$^+$CD44$^{hi}$ are shown as PSGL-1 versus CD62L. PSGL-1 low and CD62L low cells were gated as extrafollicular CD4$^+$ T cells. Percentages of PSGL-1$^{lo}$CD62L$^{lo}$ cells among CD4$^+$CD44$^{hi}$ cells are shown as mean $\pm$ SE ($n = 8$). **c** Twenty-one days after HCT, splenocytes from no-GVHD or cGVHD recipients given wild-type C57BL/6 transplants were harvested and stained for CD4, CD44, PSGL-1, and CD62L. CD44$^{hi}$CD62L$^{lo}$PSGL-1$^{lo}$CD4$^+$ T cells were sorted and used for RNA isolation and RNA-Seq microarray analysis. Heat maps of RNA expression of CXCR4, CXCR5, and CCR7 are shown as mean centered log$_2$ expression. RNA-Seq microarray measurements were performed on duplicate samples from no-GVHD group and cGVHD group. Each sample represents splenocytes from eight recipients. **d** Twenty-one days after HCT, sorted CD4$^+$CD44$^{hi}$PSGL-1$^{lo}$CD62L$^{lo}$ cells were stimulated with PMA and ionomycin for 24 h. Stimulated cells were stained and are shown as CD4 versus IFN-$\gamma$, IL-13, IL-17, or IL-21. Percentages of CD4$^+$IFN-$\gamma^+$, CD4$^+$IL-13$^+$, CD4$^+$IL-17$^+$, or CD4$^+$IL-21$^+$ cells among CD4$^+$ T cells are shown as mean $\pm$ SE ($n = 9$). *$P < 0.05$, ***$P < 0.001$, unpaired two-tailed Student's $t$ test

B-BCL6$^{-/-}$–cGVHD), and ~60% died by 60 days after HCT. We found no significant differences between the two groups, as judged by clinical manifestations (Fig. 2a–d), histopathology (Supplementary Fig. 8), or thymus damage (Supplementary Fig. 9).

Lymphoid follicles and GC area and numbers were also measured by immunofluorescent staining 60 days after HCT. B-BCL6$^{+/+}$–no-GVHD recipients had intact lymphoid follicles and GCs, while B-BCL6$^{-/-}$–no-GVHD recipients had lymphoid follicles without GCs, as expected (Fig. 2e, *top rows*). Lymphoid follicles were damaged in both B-BCL6$^{+/+}$–cGVHD and B-BCL6$^{-/-}$–cGVHD recipients, and no GCs were detected in either recipient group (Fig. 2e, *lower rows*). Consistent with the well-preserved lymphoid follicles in both B-BCL6$^{+/+}$ and B-BCL6$^{-/-}$–no-cGVHD recipients, we detected comparable and abundant percentages of T2 B cells (CD23$^+$IgD$^{hi}$IgM$^{hi}$CD21$^+$), follicular B cells (CD23$^+$IgD$^{hi}$IgM$^{lo}$CD21$^+$), and T1/marginal zone B cells (CD23$^-$IgD$^{lo}$IgM$^{hi}$) by flow cytometry (Supplementary Fig. 10A, B). In a striking contrast, B-cell subsets were barely detectable in either B-BCL6$^{+/+}$–cGVHD or B-BCL6$^{-/-}$–cGVHD recipients (Supplementary Fig. 10A, B).

To further validate the absence of GCs observed with immunofluorescent staining, we measured the percentage of Tfh and GC B cells in the spleen of recipients by using flow cytometry. Tfh cells were identified as CD4$^+$CD19$^-$PD-1$^{hi}$CXCR5$^{hi}$, and GC B cells were identified as CD19$^+$Fas$^+$GL7$^+$, as previously described[28, 37]. Consistently, BCL6 deficiency in B cells led to ~10-fold reduction in the percentage of Tfh and GC B cells in B-BCL6$^{-/-}$–no-GVHD recipients as compared to B-BCL6$^{+/+}$–no-GVHD recipients (Fig. 2f, g). GVHD caused ~65% reduction in the percentage of Tfh and GC B cells in B-BCL6$^{+/+}$–cGVHD recipients as compared to B-BCL6$^{+/+}$–no-GVHD recipients, although the percentages of these cells were still significantly higher than in B-BCL6$^{-/-}$–cGVHD or B-BCL6$^{-/-}$–no-GVHD recipients (Fig. 2f, g). Taken together, these results indicate that GC formation is not required for induction of cGVHD and its associated lymphopenia.

**Chronic GVHD is linked with PSGL-1$^{lo}$CD4$^+$ T-cell expansion.** We explored how extrafollicular CD4$^+$ T- and B-cell interaction could mediate cGVHD. PSGL-1 has been recently described as a new immune check point for T cells[41], and extrafollicular CD4$^+$ T cells in autoimmune mice have decreased expression of PSGL-1[42]. Extrafollicular CD4$^+$ T cells were identified as CD44$^{hi}$CD62$^{lo}$PSGL-1$^{lo}$CD4$^+$ (PSGL-1$^{lo}$CD4$^+$) T cells with high-level expression of ICOS, increased expression of CXCR4, and decreased expression of CXCR5, while T$_{FH}$ had increased expression of CXCR5. PSGL-1$^{lo}$CD4$^+$ T cells also had increased production of IFN-γ, IL-13, and IL-21[25]. Because expansion of PSGL-1$^{lo}$CD4$^+$ T cells was observed in mice with autoimmune lupus[25], we tested whether this population was also expanded during the pathogenesis of cGVHD. In our model, evidence of cutaneous GVHD begins at ~20 days after HCT and develops rapidly between days 30– 45 after HCT (Figs. 1a and 2a). Therefore, we measured the percentage of PSGL-1$^{lo}$CD4$^+$ T cells among total donor CD4$^+$ T cells in the spleen, lung, and liver on days 21, 30, and 45 after HCT.

WT C57BL/6 donor grafts were transplanted into lethally irradiated BALB/c recipients as described in Fig. 1. As compared with no-cGVHD recipients given TCD-BM cells, cGVHD recipients given spleen cells had ~5-fold higher percentages of PSGL-1$^{lo}$CD4$^+$ T cells in the spleen, lung, and liver on day 21 (Fig. 3a, b and Supplementary Fig. 11). The difference subsequently diminished in the spleen but was maintained in the lung and liver on day 30 and then disappeared by day 45 after HCT (Fig. 3a, b and Supplementary Fig. 11). The expanded

PSGL-1$^{lo}$CD4$^+$ T cells from the spleen of cGVHD recipients had slightly increased expression of CXCR4 but obviously decreased expression of CXCR5 and CCR7, as measured by RNA-seq analysis and suitably confirmed by real-time PCR (Fig. 3c and Supplementary Fig. 12). They also had 3–5-fold higher percentages of IFN-γ$^+$, IL-13$^+$, IL-17$^+$, and IL-21$^+$ cells (Fig. 3d). On the other hand, even at the peak expansion time of PSGL-1$^{lo}$CD4$^+$ cells, day 21 after HCT, there was no obvious expansion of Tfh cells in the spleen of cGVHD recipients (Supplementary Fig. 13). These results indicate that cGVHD development is associated with expansion of PSGL-1$^{lo}$CD4$^+$ T cells but not Tfh cells.

In addition, expansion of PSGL-1$^{lo}$CD4$^+$ T cells was also observed in MHC-matched LP/J (H-2$^{bc}$) donor → C57BL/6 (H-2$^b$) model. Compared with no-cGVHD recipients given TCD-BM, cGVHD recipients given additional donor spleen cells had ~10-fold increase in percentages of PSGL-1$^{lo}$CD4$^+$ T cells in the spleen of recipients at 21 days after transplantation (Supplementary Fig. 14A). In contrast, there was no expansion of PSGL-1$^{lo}$CD4$^+$ T cells in C57BL/6 recipients given syngeneic transplants (Supplementary Fig. 14B). These results indicate that expansion of PSGL-1$^{lo}$CD4$^+$ T cells in cGVHD recipients is driven by MHC-matched and mismatched alloimmune responses but not by homeostatic expansion.

**Blocking ICOS/ICOSL binding reduces PSGL-1$^{lo}$CD4$^+$ T expansion.** ICOS-dependent expansion of extrafollicular PSGL-1$^{lo}$CD4$^+$ T cells is important in autoimmune pathogenesis in MRL$^{lpr}$ mice that have defective GC formation[25]. By RNA-seq analysis, we found that PSGL-1$^{lo}$CD4$^+$ T cells in cGVHD recipients with destruction of GCs upregulated expression of ICOS, although they downregulated expression of other costimulatory and coinhibitory molecules such as PD-1, CD80, and PD-L1 (Fig. 4a). The upregulation of ICOS and downregulation of PD-1, CD80, and PD-L1 were suitably confirmed by real-time PCR (Supplementary Fig. 15). We tested whether blockade of ICOS and ICOS-ligand interaction prevented expansion of PSGL-1$^{lo}$CD4$^+$ T and induction of cGVHD in recipients given B-BCL6$^{-/-}$ grafts, because these recipients have only extrafollicular CD4$^+$ T and B interactions but no follicular GC formation. Accordingly, lethally irradiated BALB/c recipients were transplanted with spleen and TCD-BM from B-BCL6$^{-/-}$ C57BL/6 donors and treated with blocking anti-ICOS mAb or control rat IgG2b (200 μg/mouse), every other day from days 0 to 45 after HCT. Recipients given TCD-BM alone were used as no-cGVHD controls. ICOS blockade markedly ameliorated cGVHD severity as judged by clinical score, survival, and histopathology (Fig. 4b–f).

As compared to cGVHD recipients treated with control rat IgG2b, recipients with cGVHD prevented by ICOS blockade had a marked reduction in the percentage of PSGL-1$^{lo}$CD4$^+$ T cells in the spleen, lung, and liver (Fig. 5a), which was associated with reduction of serum anti-dsDNA-IgG concentrations (Fig. 5b). In addition, we tested whether the reduced percentages of PSGL-1$^{lo}$CD4$^+$ T cells were associated with specific blockade of ICOS and ICOS-ligand interaction. As compared with CD4$^+$ T cells from recipients treated with control rat IgG2b, the CD4$^+$ T cells from recipients treated with anti-ICOS mAb had reduced direct anti-ICOS staining with increased anti-rat IgG2b staining (Fig. 5c, d), indicating that anti-ICOS (IgG2b) mAb bound to the ICOS expressing CD4$^+$ T cells and blocked anti-ICOS staining. Furthermore, anti-ICOS mAb treatment led to upregulated expression of ICOS-L by B cells, indicating disruption of ICOS–ICOSL interaction between CD4$^+$ T and B cells (Fig. 5e). Taken together, these results indicate that in recipients without GC formation, blockade of ICOS/ICOSL interaction reduces

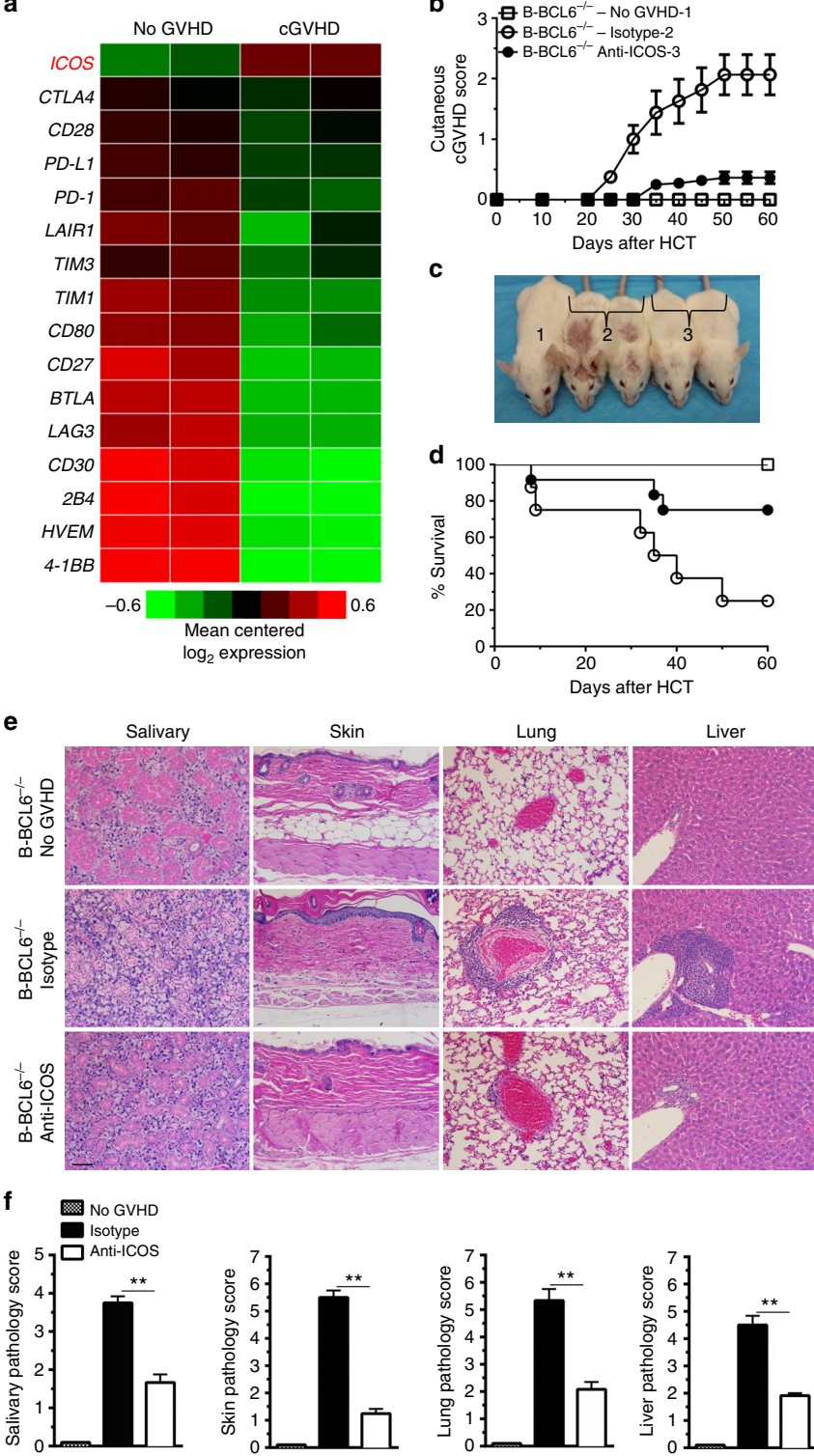

**Fig. 4** Anti-ICOS treatment ameliorates cGVHD in recipients without GC formation. **a** As described in Fig. 3, 21 days after HCT, splenocytes from no-GVHD or cGVHD recipients given wild-type C57BL/6 donor cells were harvested and stained for CD4, CD44, PSGL-1, and CD62L. CD44$^{hi}$CD62L$^{lo}$PSGL-1$^{lo}$CD4$^{+}$ T cells were sorted and used for RNA isolation and RNA-Seq microarray analysis. Heat maps of RNA expression levels of costimulatory and coinhibitory markers are shown as mean centered log$_2$ expression. RNA-Seq microarray measurements were performed on duplicate samples from no-GVHD group and cGVHD group. Each sample represents splenocytes from eight recipients. **b–f** BALB/c recipients were irradiated (850 cGy) and given either 2.5 × 10$^6$ TCD-BM ($n = 8$) or 2.5 × 10$^6$ TCD-BM plus 1 × 10$^6$ splenocytes ($n = 12$) from B-BCL6$^{-/-}$ C57BL/6 donors. Recipients given 2.5 × 10$^6$ TCD-BM plus 1 × 10$^6$ splenocytes were treated with anti-ICOS or isotype control of rat IgG2b, 200 μg/mouse i.p., starting on day 0 and repeated every other day until day 45 after HCT. Chronic GVHD was monitored. **b** Cutaneous cGVHD score (Group 3 versus Group 2: $P < 0.001$ two-way ANOVA). **c** Picture taken on day 60 after HCT(1–no-GVHD, 2–isotype, 3–anti-ICOS). **d** Survival curve (Group 3 versus Group 2: $P < 0.05$, log-rank test). **e** H&E staining of salivary gland, skin, lung, and liver. **f** Pathology scores of cGVHD for salivary gland, skin, lung, and liver are shown as mean ± SE ($n = 6$). **$P < 0.01$, unpaired two-tailed Student's $t$ test. *Scale bar*, 50 μm

extrafollicular PSGL-1$^{lo}$CD4$^+$ T-cell expansion, reduces autoantibody production, and ameliorates cGVHD. These results suggest that extrafollicular PSGL-1$^{lo}$CD4$^+$ T cells have a critical function in the pathogenesis of cGVHD.

**Donor BCL6 deficiency reduces PSGL-1$^{lo}$CD4$^+$ T expansion.** CD4$^+$ T-cell expression of BCL6 is necessary for extrafollicular CD4$^+$ T- and B-cell interactions in non-autoimmune mice[19]. Therefore, we tested the impact of BCL6 deficiency in donor

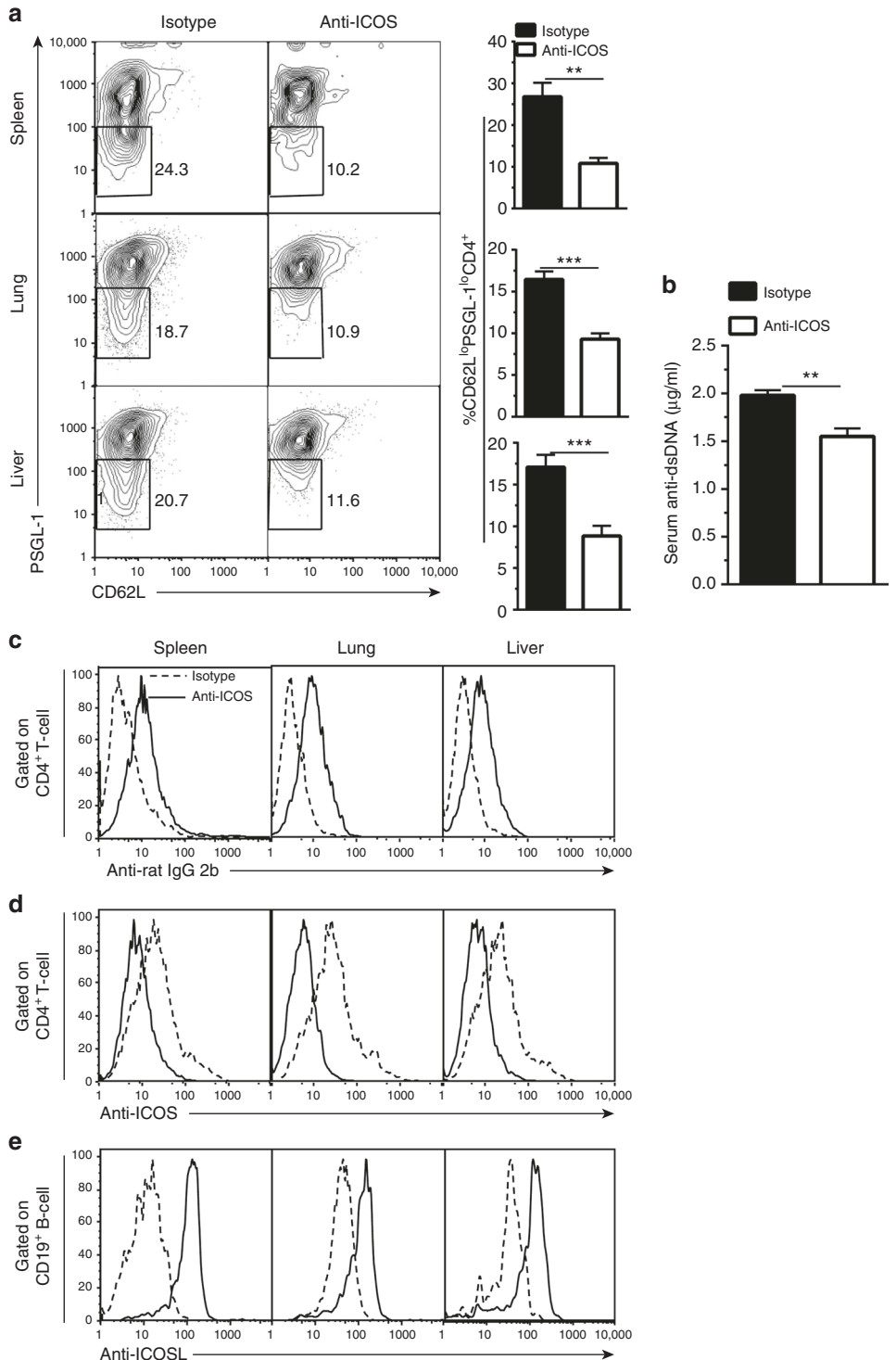

**Fig. 5** Anti-ICOS treatment reduces PSGL-1$^{lo}$CD4$^+$ T-cell expansion. As described in Fig. 4, BALB/c recipients were irradiated (850 cGy) and given 2.5 × 10$^6$ TCD-BM plus 1 × 10$^6$ splenocytes. Recipients were treated with anti-ICOS or control rat IgG2b (200 μg/mouse i.p.) every other day from day 0 to day 45 after HCT. **a** Twenty-one days after HCT, mononuclear cells from spleen, lung, and liver were stained for CD4, CD44, PSGL-1, and CD62L. Gated CD4$^+$CD44$^{hi}$ cells are shown as PSGL-1 versus CD62L ($n = 8$). **b** Serum anti-dsDNA was measured at 45 days after HCT. **c–e** Twenty-one days after HCT, mononuclear cells from spleen, lung, and liver were stained with CD4, anti-rat IgG2b, or anti-ICOS, or stained with anti-CD19 and anti-ICOSL. Gated CD4$^+$ T cells are shown as anti-rat IgG2b (**c**) or anti-ICOS (**d**) staining. **e** Gated CD19$^+$ B cells are shown as ICOSL staining. One representative of four experiments is shown. **P < 0.01, ***P < 0.001, unpaired two-tailed Student's *t* test

CD4$^+$ T cells on induction of autoimmune-like cGVHD and expansion of PSGL-1$^{lo}$CD4$^+$ T cells. Accordingly, TCD-BM ($2.5 \times 10^6$) and spleen cells ($1 \times 10^6$) from BCL6$^{fl/fl}$ CD4-Cre-C57BL/6 (CD4-BCL6$^{-/-}$) donors that have BCL6 deficiency specifically in CD4$^+$ T cells or from control BCL6$^{fl/fl}$ C57BL/6 (CD4-BCL6$^{+/+}$) littermates were transplanted into lethally irradiated BALB/c recipients. Recipients given TCD-BM alone from either donor showed no signs of GVHD (no-GVHD) and were combined into a single control group. Recipients given TCD-BM and spleen cells from control CD4-BCL6$^{+/+}$ donors developed severe cGVHD with weight loss and hair loss, and ~75% died within 60 days after HCT (Fig. 6a–d). In contrast, recipients given TCD-BM and spleen cells from CD4-BCL6$^{-/-}$ donors had weight loss and ~30% mortality but no signs of cutaneous cGVHD (Fig. 6a–d). In addition, CD4-BCL6$^{-/-}$–cGVHD recipients showed little skin tissue damage by histopathology and had reduced damage in the thymus as indicated by higher numbers of CD4$^+$CD8$^+$ thymocytes, but the extent of salivary gland damage was similar as compared to CD4-BCL6$^{+/+}$–cGVHD recipients (Supplementary Fig. 16). These results indicate that BCL6 deficiency in donor CD4$^+$ T cells prevents cGVHD-associated damage in the skin and decreases damage in the thymus.

In the spleen of recipients given CD4-BCL6$^{-/-}$ TCD-BM, GCs were not completely absent, but their size and numbers were decreased by more than 85% as compared to recipients given CD4-BCL6$^{+/+}$ TCD-BM (Fig. 6e). CD4-BCL6$^{+/+}$–cGVHD recipients had no lymphoid follicles or GCs, while CD4-BCL6$^{-/-}$–cGVHD recipients had lymphoid follicles but no GCs (Fig. 6e). Compared with CD4-BCL6$^{+/+}$–cGVHD recipients, CD4-BCL6$^{-/-}$–cGVHD recipients had lower numbers of PSGL-1$^{lo}$CD4$^+$ T cells in the spleen, lung, and liver (Fig. 6f).

To further test the role of PSGL1$^{lo}$CD4$^+$ T cells in skin and thymus damage in cGVHD recipients, sorted PSGL1$^{lo}$CD4$^+$ T cells ($1 \times 10^6$) from the spleen of cGVHD recipients given B-BCL6$^{-/-}$ transplants at 21 days after HCT were injected into recipients given CD4-BCL6$^{-/-}$ transplants. The control recipients were given transplantation buffer only. Injection of the PSGL1$^{lo}$CD4$^+$ T cells augmented clinical signs of cutaneous GVHD and skin pathology and exacerbated loss of CD4$^+$CD8$^+$ thymocytes (Supplementary Fig. 17). These results indicate that expansion of extrafollicular PSGL-1$^{lo}$CD4$^+$ T cells in cGVHD recipients is BCL6-dependent and that prevention of extrafollicular CD4$^+$ T-cell expansion can effectively prevent induction of cutaneous cGVHD, although other manifestations of cGVHD persist.

**Donor Stat3 deficiency reduces PSGL-1$^{lo}$CD4$^+$ T expansion**. Transcription factors, Stat3 and BCL6, are both expressed by pre-follicular CD4$^+$ T cells and are required for differentiation of Tfh and GC formation[14, 39]. Therefore, we used RNA-seq analysis to compare the expression levels of these transcription factors in PSGL-1$^{lo}$CD4$^+$ T cells from the spleen of no-GVHD recipients given WT-TCD-BM and cGVHD recipients given WT-TCD-BM and spleen cells. Although BCL6 was required for expansion of PSGL-1$^{lo}$CD4$^+$ T cells in cGVHD recipients (Fig. 6), expression of BCL6 appeared to be downregulated in those cells (Fig. 7a). However, expression of Stat3 in those cells was slightly increased although not significant, as measured by RNA-seq and suitably confirmed by real-time PCR analysis (Fig. 7a and Supplementary Fig. 18). These data suggested a possible function of Stat3 in maintaining the expansion of PSGL-1$^{lo}$CD4$^+$ T cells and the induction of cGVHD. Therefore, we evaluated PSGL-1$^{lo}$CD4$^+$ T-cell expansion and cGVHD induction by Stat3-deficient donor CD4$^+$ T cells.

Accordingly, TCD-BM ($2.5 \times 10^6$) and spleen cells ($1 \times 10^6$) from Stat3$^{fl/fl}$ CD4-Cre-C57BL/6 donors that have Stat3

deficiency specifically in CD4$^+$ T cells (CD4-Stat3$^{-/-}$) or from control Stat3$^{fl/fl}$ C57BL/6 (CD4-Stat3$^{+/+}$) littermates were transplanted into lethally irradiated BALB/c recipients. Recipients given CD4-Stat3$^{-/-}$ transplants had no clinical evidence of cGVHD in the skin and no mortality beyond day 15 after HCT (Fig. 7b–e). Prevention of cGVHD in recipients given CD4-Stat3$^{-/-}$ grafts was confirmed by lack of damage in the thymus, as judged by normal percentages of CD4$^+$CD8$^+$ thymocytes and the lack of damage in the skin and salivary glands (Supplementary Fig. 19). Although Stat3$^{-/-}$ PSGL-1$^{lo}$CD4$^+$ T cells expressed similar levels of BCL6 as compared to wild-type controls (Supplementary Fig. 20), prevention of cGVHD in recipients given Stat3$^{-/-}$ transplants was associated with lower percentages of PSGL-1$^{lo}$CD4$^+$ T cells in the spleen, lung, and liver tissues (Fig. 7f and Supplementary Fig. 21). The prevention of cGVHD was also associated with lower serum concentrations of anti-dsDNA-IgG (Supplementary Fig. 22) and less IgG deposition in the skin and thymus (Supplementary Fig. 23A, B). In addition, recipients given CD4-Stat3$^{-/-}$ grafts had increased percentages of Foxp3$^+$CD4$^+$ Treg cells as compared to cGVHD recipients given WT grafts or CD4-BCL6$^{-/-}$ grafts ($P < 0.001$, Supplementary Fig. 24).

GC formation in recipients given CD4-Stat3$^{-/-}$ TCD-BM was not completely absent, but the size and numbers of GCs were reduced by more than 50% (Fig. 7g) as compared to no-GVHD recipients given CD4-Stat3$^{+/+}$ TCD-BM. Chronic GVHD recipients given CD4-Stat3$^{+/+}$ TCD-BM and spleen cells completely lost lymphoid follicles and GCs (Fig. 7g). In contrast, cGVHD-free recipients given CD4-Stat3$^{-/-}$ TCD-BM and spleen cells had preserved lymphoid follicles, and their GC size and numbers were similar to those in no-GVHD recipients given CD4-Stat3$^{-/-}$ TCD-BM alone (Fig. 7g).

We further evaluated the function of PSGL-1$^{lo}$CD4$^+$ T cells in thymus damage early after HCT in recipients given Stat3$^{+/+}$ or Stat3$^{-/-}$ transplants, because CD4$^+$ T cells are important in damaging thymus via expression of FasL early after allogenic HCT[43]. Moreover, thymus damage is important in persistence of chronic GVHD[7, 9]. First, we measured thymus damage with immunofluorescent staining of thymic cortical (CK8) and medullary (UEA-1) epithelial cells and percentage of CD4$^+$CD8$^+$ thymocytes at 10 and 30 days after HCT. We observed that thymus damage was similar in recipients given CD4-Stat3$^{+/+}$ or CD4-Stat3$^{-/-}$ transplants at 10 days after HCT, with loss of UEA-1$^+$ medullary epithelial cells and CD4$^+$CD8$^+$ thymocytes (Fig. 8a). By 30 days after HCT, the difference become obvious, and the recovery of cortical and medullary epithelial cells and CD4$^+$CD8$^+$ thymocytes in recipients given Stat3$^{-/-}$ transplants was much better than in recipients given Stat3$^{+/+}$ transplants (Fig. 8b). The better thymic recovery in recipients given Stat3$^{-/-}$ transplants was associated with much lower numbers of PSGL-1$^{lo}$CD4$^+$ T cells in the spleen and marked reduction of infiltration in the thymus (Fig. 8c). The PSGL-1$^{lo}$CD4$^+$ T cells in the spleen of recipients also expressed lower levels of CCR9 and FasL (Fig. 8d). CCR9 enables CD4$^+$ T migration into thymus[44] and FasL expressed by the CD4$^+$ T cells can mediate thymus damage[43].

Taken together, these results indicate that deficiency of Stat3 in donor CD4$^+$ T cells results in (1) marked reduction of expansion of extrafollicular PSGL-1$^{lo}$CD4$^+$ T cells and reduction of their infiltration of thymus; (2) lower production of serum IgG autoantibodies and decreased IgG deposition in GVHD target tissues such as skin and thymus. Finally, reduction of PSGL-1$^{lo}$CD4$^+$ T infiltration and decreased IgG deposition allow recovery of the thymus early after HCT with production of Foxp3$^+$ Treg cells, leading to full prevention of cGVHD.

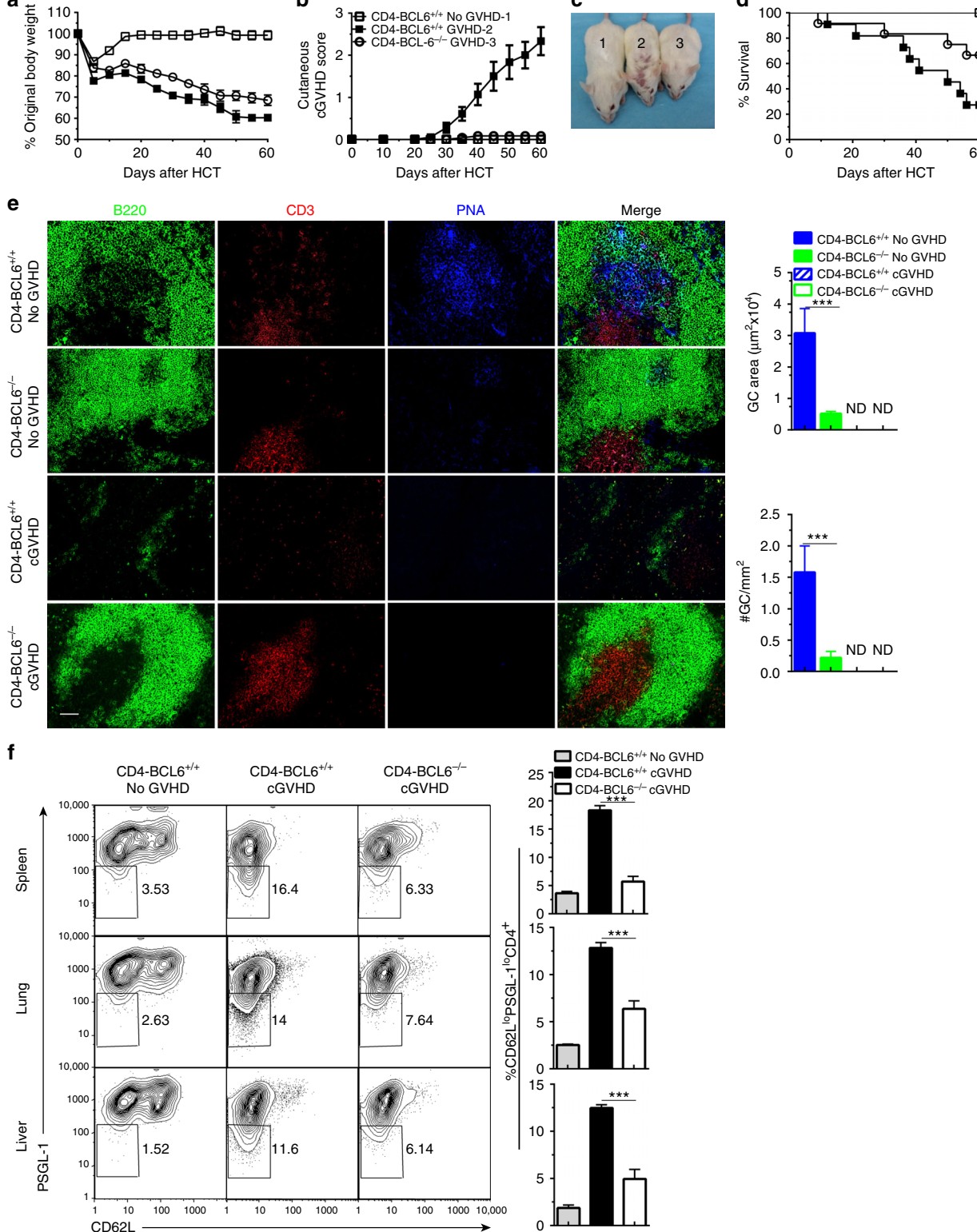

**Fig. 6** BCL6 deficiency in donor CD4$^+$ T cells prevents expansion of PSGL-1$^{lo}$CD4$^+$ T cells and cutaneous cGVHD. BALB/c recipients were irradiated (850 cGy) and given $2.5 \times 10^6$ TCD-BM alone or $2.5 \times 10^6$ TCD-BM plus $1 \times 10^6$ splenocytes from either WT or B-BCL6$^{-/-}$ C57BL/6 donors ($n = 12$). cGVHD was monitored. **a** Percent body weight changes. **b** Cutaneous cGVHD scores (Group 3 versus Group 2: $P < 0.001$ two-way ANOVA). **c** Picture taken at day 60 after HCT (1 and 2–CD4-BCL6$^{+/+}$ no GVHD and GVHD, 3–CD4-BCL6$^{-/-}$ GVHD). **d** Survival curve. **e** Sixty days after transplantation, spleens were harvested, and germinal centers were identified by immunofluorescent staining of B220, CD3, and PNA. GC area and numbers were measured and are shown as mean ± SE ($n = 4$). **f** Twenty-one days after HCT, mononuclear cells from spleen, lung, and liver were stained for CD4, CD44, PSGL-1, and CD62L. Gated CD4$^+$CD44$^{hi}$ cells are shown as PSGL-1 versus CD62L. Percentages of CD62L$^{lo}$PSGL-1$^{lo}$ cells among CD4$^+$CD44$^{hi}$ are shown as mean ± SE ($n = 8$). ***$P < 0.001$, unpaired two-tailed Student's $t$ test. CD4-BCL6$^{+/+}$ no-GVHD = CD4-BCL6$^{+/+}$ TCD-BM; CD4-BCL6$^{-/-}$ no-GVHD = CD4-BCL6$^{-/-}$ TCD-BM; CD4-BCL6$^{+/+}$ cGVHD = CD4-BCL6$^{+/+}$ TCD-BM + CD4-BCL6$^{+/+}$ splenocytes; CD4-BCL6$^{-/-}$ cGVHD = CD4-BCL6$^{-/-}$ TCD-BM + CD4-BCL6$^{-/-}$ splenocytes. *Scale bar*, 50 μm

## Discussion

The respective functions of extrafollicular and follicular CD4+ T–B interactions in cGVHD pathogenesis have remained unclear and controversial. By using different mouse models of cGVHD and donors with BCL6 deficiency specifically in B cells or CD4+ T cells and Stat3 deficiency in CD4+ T cells, we have demonstrated that (1) GC formation and follicular CD4+ T–B interaction is dispensable for induction of cGVHD, and extrafollicular CD4+ T- and B-cell interactions are sufficient for induction of cGVHD; (2) cGVHD pathogenesis is associated with expansion of PSGL-1loCD4+ T cells. PSGL-1loCD4+ T cells express high levels of ICOS and IL-21, and their expansion is Stat3- and BCL6-dependent; (3) BCL6 deficiency in CD4+ T cells prevents expansion of PSGL-1loCD4+ T cells and partially prevents systemic cGVHD, with effective prevention of cutaneous cGVHD; (4) Stat3 deficiency in donor CD4+ T cells prevents expansion of PSGL-1loCD4+ T cells and allows full recovery of the thymus with production of Foxp3+ Treg cells, fully preventing induction of cGVHD.

We have demonstrated that GC formation is not required for induction of cGVHD. We showed that although bone marrow transplants with BCL6 deficiency specifically in B cells (B-BCL6−/−) were not able to form GCs, spleen cells from B-BCL6−/− donors and WT donors induced similar severity of cGVHD. This observation is consistent with the observations that cGVHD in animal models and humans is often associated with lymphopenia at the disease onset[9–11, 35]. This observation is also consistent with the previous studies showing that allogeneic HCT patients did not have immunoglobulin somatic hypermutation, a process that requires GC formation[33, 34]. This observation is also consistent with results of a recent study showing that patients with active cGVHD had low numbers of Tfh cells in the blood[31]. Our observations differ from those in previous reports showing that cGVHD in murine models was associated with enlarged GCs and that GC formation was required for induction of cGVHD[28–30]. Explanations for the different results remain unclear.

Lymphopenia in cGVHD recipients results from damage to lymphoid niches[11]. Destruction or loss of GC formation in cGVHD may also result from damage to lymphoid niches during GVHD development. Follicular DCs and lymphotoxin-producing stromal cells represent important lymphoid niches[45]. Our recent publications[5, 9] indicate that alloreactive and autoreactive CD4+ T cells and antibodies from donor B cells contribute to destruction of B-cell follicles, GCs, and follicular DCs during chronic GVHD development.

We found that development of cGVHD is associated with expansion of PSGL-1loCD4+ T cells and with reduced percentages or loss of Tfh and GC B cells in the spleen. While cGVHD recipients showed loss of GCs and Tfh cells at the disease onset ~21 days after HCT, recipients showed high percentages of PSGL-1loCD4+ T cells in the spleen, liver, and lung. The percentages of PSGL-1loCD4+ T cells declined to control levels at the peak time of cGVHD, ~45 days after HCT. PSGL-1loCD4+ T cells expressed high levels of ICOS, IL-21, IL-13, and IL-17. Blockade of ICOS signaling effectively prevented their expansion and effectively ameliorated cGVHD. In addition, add-back of PSGLloCD4+ T cells to recipients given CD4-BCL6−/− transplants that cannot give rise to PSGL1loCD4+ T cells resulted in marked augmentation of damage in the thymus and development of cutaneous cGVHD. These results indicate that PSGLloCD4+ T cells are important in the pathogenesis of cGVHD.

Our observations are consistent with previous reports that blockade of ICOS/ICOS-ligand interaction ameliorated acute and chronic GVHD[28, 46–48]. On the other hand, our experiments have unraveled the role of ICOS/ICOS-ligand interaction in augmenting expansion of extrafollicular PSGL-1loCD4+ T cells. Since there was an absence of GC in cGVHD recipients, amelioration of cGVHD by blockade of ICOS/ICOS-ligand interaction described in a previous publication[28] may not result from reduction of GC formation, but may result from blockade of interactions between extrafollicular CD4+ T cells and B cells. This conclusion is consistent with observations that development of lupus in MRLlpr mice is associated with expansion of PSGL-1loCD4+ T cells but not with expansion of Tfh cells[25].

Our previous studies indicate that CD4+ T and B interactions and antibody production are required to perpetuate cGVHD and augment Th17 infiltration of skin tissues[6, 9], suggesting that CD4+ T- and B-cell interaction may continue in GVHD target tissues, even after destruction of lymphatic tissues. This suggestion is consistent with our observation that the percentages of PSGL-1loCD4+ T cells in the spleen decreased shortly after the onset of cGVHD but persisted much longer in GVHD target tissues. Local T/B interactions in inflamed tissues is supported by non-Tfh CXCR5−CD4+ helper T cells[49, 50]. Thus, we hypothesize that PSGL-1loCD4+ T cells and B cells may interact in GVHD target tissues in the pathogenesis of cGVHD. Further studies are needed to characterize the surface phenotype and function of PSGL-1loCD4+ T cells at different target tissues at different stages of cGVHD development, and to determine how PSGL-1loCD4+ T cells interact with B cells and other Th subsets during the pathogenesis of cGVHD.

Early after activation, CD4+ T cells downregulate CCR7 and PSGL-1 and migrate to T-B border; some of the CD4+ T cells become pre-Tfh under influence of cytokines (such as IL-6 and IL-21), with upregulation of Stat3 and BCL6 as well as CXCR5 and ICOS. Pre-Tfh CD4+ T cells can interact with B cells at the T–B border to produce IgG antibodies[14, 28]. We observed that, like pre-Tfh cells, PSGL-1loCD4+ T cells in cGVHD are also Stat3 and BCL6 dependent and express high levels of ICOS and IL-21. This observation indicates that PSGL-1loCD4+ T cells most likely develop at the T–B border during the pre-Tfh stage of development. Unlike pre-Tfh cells, however, PSGL-1loCD4+ T cells in cGVHD recipients express low levels of CXCR5. PSGL-1loCD4+ T cells in cGVHD recipients could be a subset of pre-Tfh cells, because pre-Tfh cells are heterogeneous, and some of them are CXCR5lo[28].

Consistently, we and others have observed that high levels of IL-6 are associated with onset of cGVHD[35, 51, 52]. Mechanisms that drive expansion of PSGL-1loCD4+ T cells and the reasons why this expansion does not lead to formation of Tfh or GCs remain unclear. We hypothesize that GVHD-related alloimmunity damages the host-type follicular DC and stromal cells in lymphoid tissues early after HCT, before donor-type follicular DC and stromal cells join the lymphoid structure. This mechanism would explain why donor-type CD4+ T and B cells can interact at T–B border but cannot move forward to form intact lymphoid follicles or GCs.

Our recent study showed that IgG antibody production in cGVHD recipients is associated with destruction of lymphoid tissues[9], suggesting that IgG antibodies resulting from extrafollicular CD4+ T and B interactions may augment damage to lymphoid structures, which can also prevent GC formation. Recipients given CD4-BCL6−/− CD4+ T cells had lymphoid follicles but did not have GCs, unlike the destruction of lymphoid follicles in recipients given WT CD4+ T cells with cGVHD. This observation further supports our hypothesis that the interaction of autoreactive PSGL-1loCD4+ T cells with B cells and the production of IgG autoantibodies are important in damaging lymphoid structures.

Stat3 deficiency in donor CD4+ T cells fully prevented induction of cGVHD, but BCL6 deficiency in donor CD4+ T cells only partially prevented cGVHD, although both effectively prevented

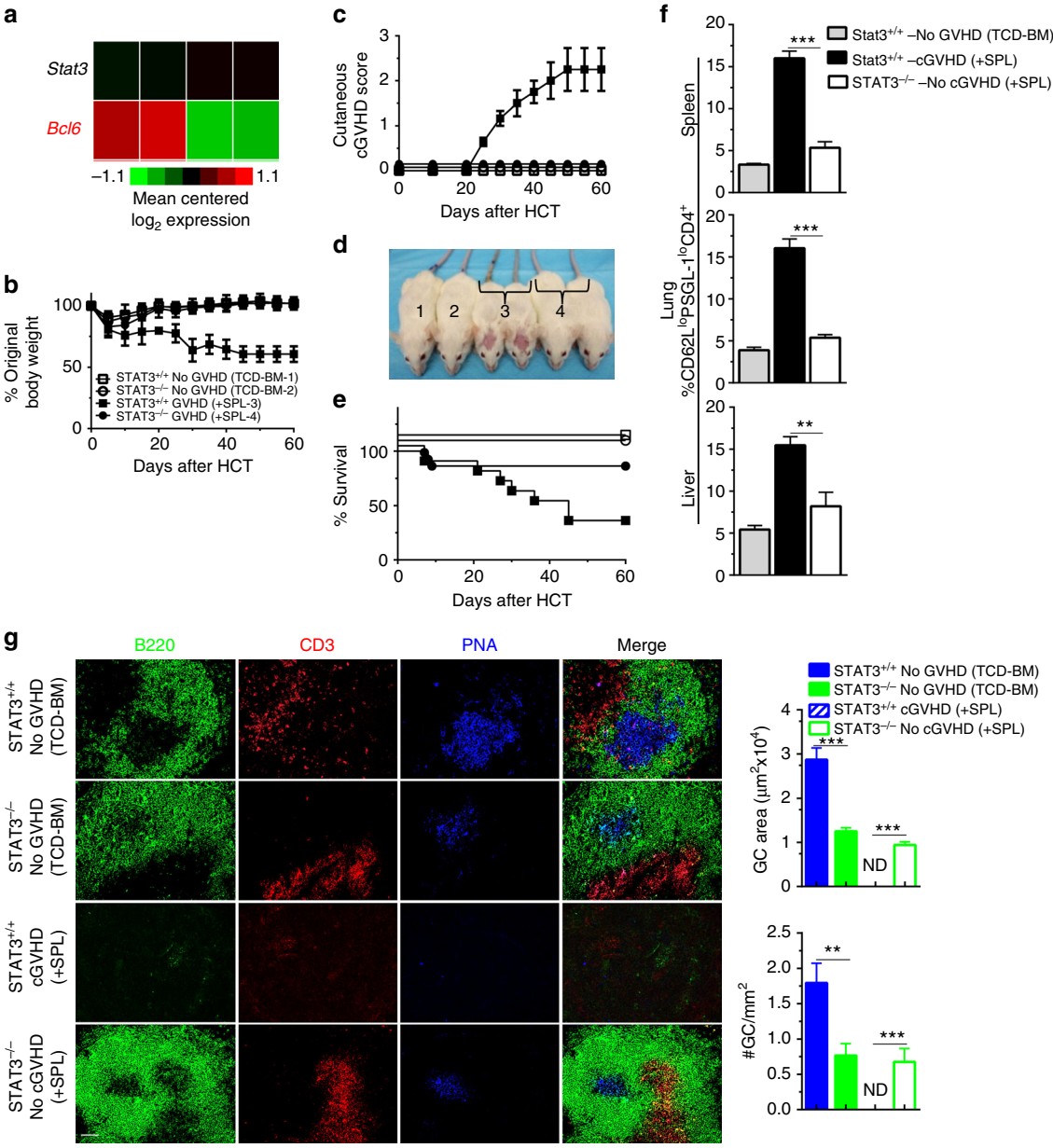

**Fig. 7** Stat3 deficiency in donor CD4+ T cells prevents expansion of PSGL-1loCD4+ T cells and systemic cGVHD. **a** Twenty-one days after HCT, splenocytes from no-GVHD or cGVHD recipients given wild-type C57BL/6 donors were harvested and stained for CD4, CD44, PSGL-1, and CD62L. CD44hiCD62LloPSGL-1loCD4+ T cells were sorted and used for RNA isolation and RNA-Seq microarray analysis. Heat maps of RNA expression of transcription factor in extrafollicular T cells are shown as mean centered log2 expression. RNA-Seq microarray measurements were performed on duplicate samples from no-GVHD and cGVHD groups. Each sample represents splenocytes from eight recipients. **b–g** BALB/c recipients were irradiated (850 cGy) and given $2.5 \times 10^6$ TCD-BM alone or $2.5 \times 10^6$ TCD-BM plus $1 \times 10^6$ splenocytes ($n = 12$) from either WT or CD4-STAT3−/− C57BL/6 donors. **b** Percent body weight changes (Group 3 versus Group 4: $P < 0.001$, two-way ANOVA). **c** Cutaneous cGVHD scores (Group 3 versus Group 4: $P < 0.001$, two-way ANOVA). **d** Picture taken on day 60 after HCT (1 and 3–Stat3+/+–no GVHD and cGVHD, 2 and 4–Stat3−/−–no GVHD and cGVHD). **e** Survival curve (Group 3 versus Group 4: $P < 0.05$, log-rank test). **f** Twenty-one days after HCT, mononuclear cells from spleen, lung, and liver were stained with CD4, CD44, PSGL-1, and CD62L. Percentages of CD62LloPSGL-1lo cells among CD4+CD44hi are shown as mean ± SE ($n = 6$). **g** Sixty days after transplantation, spleens were harvested and germinal centers were identified by immunofluorescent staining of B220, CD3, and PNA, and GC area and numbers were measured and are shown as mean ± SE ($n = 4$). **$P < 0.01$***$P < 0.001$, unpaired two-tailed Student's $t$ test. CD4-Stat3+/+ no GVHD = CD4-Stat3+/+ TCD-BM; CD4-Stat3−/− no GVHD = CD4-Stat3−/−TCD-BM; CD4-Stat3+/+ cGVHD = CD4-Stat3+/+ TCD-BM + CD4-Stat3+/+ splenocytes; CD4-Stat3−/− cGVHD = CD4-Stat3−/− TCD-BM + CD4-Stat3−/− splenocytes. *Scale bar*, 50 μm

expansion of PSGL-1loCD4+ T cells. The difference may result from the ability of Stat3 to regulate development of Foxp3+ Treg and RoRγt+ Th17 cells[53, 54] in addition to BCL6+ pre-Tfh and Tfh cells[14], whereas BCL6 in CD4+ T cells regulates only the development of pre-Tfh and Tfh[39, 55]. Stat3 deficiency in donor CD4+

T cells not only reduced expansion of PSGL-1loCD4+ T cells and reduced their infiltration of thymus, but also reduced serum concentrations of anti-dsDNA autoantibody and deposition in thymus tissues, allowing for full recovery of the thymus and its production of Treg cells that control peripheral autoreactive

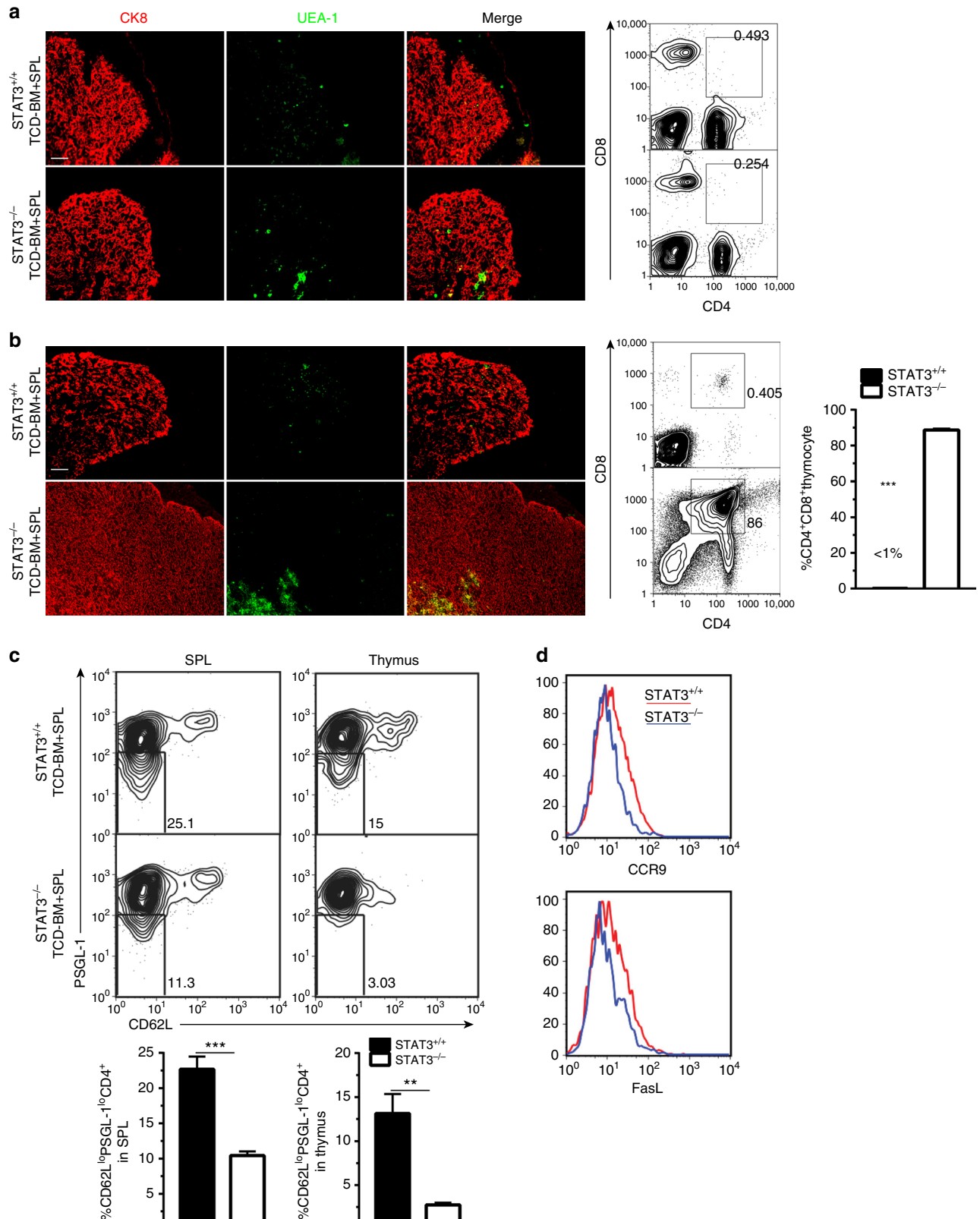

**Fig. 8** Thymus recovery in recipients given Stat3[−/−] transplants is associated with reduced PSGL-1[lo]CD4[+] T-cell infiltration in the thymus. BALB/c recipients were conditioned with 850 cGy TBI and given 2.5 × 10[6] TCD-BM plus 1 × 10[6] splenocytes from either WT or CD4-Stat3[−/−] C57BL/6 donors. **a**, **b** 10 days (**a**) and 30 days (**b**) after HCT, thymus specimens were harvested and stained with CK8 for the cortex and UEA-1 for the medulla epithelial cells. Percentage of CD4[+]CD8[+] thymocytes was measured with flow cytometry. **c** Ten days after HCT, spleen and thymus were harvested and stained for CD4, CD44, PSGL-1, and CD62L. Gated CD4[+]CD44[hi] are shown as PSGL-1 versus CD62L. PSGL-1[lo]CD62L[lo]CD4[+]CD44[hi] cells were identified as extrafollicular PSGL1[lo] CD4[+] T cells. Percentages of PSGL-1[lo] CD4[+] T cells among CD4[+]CD44[hi] cells are shown as mean ± SE (*n* = 6). **d** CCR9 and FasL expression on splenic PSGL-1[lo]CD4[+] T cells were measured by flow cytometry and one representative histogram is shown (*n* = 6). *Scale bar*, 50 μm

T-cell expansion, leading to full prevention of cGVHD development. Our observations are not only consistent with previous reports that Stat3 deficiency in donor CD4+ T cells leads to production of thymus-derived natural Treg cells that are required to prevent cGVHD[56], but also provide further evidence on explaining how Stat3 deficiency in donor CD4+ T cells allows full recovery of the thymus and production of natural Treg cells.

We and others also showed that infusion or in vivo expansion of donor-type Treg cells can ameliorate ongoing cGVHD[57, 58]. On the other hand, deletion of Stat3 and BCL6 in CD4+ T cells decreases the numbers of follicular regulatory Treg cells[59, 60]. We observed that GC were tiny or absent in recipients given TCD-BM only or TCD-BM and spleen cells from donors with specific deletion of Stat3 or BCL6 in CD4+ T cells, suggesting that those recipients have low numbers of follicular Treg cells. However, cGVHD was absent in recipients given Stat3-deficient CD4+ T cells and showed decreased severity in recipients given BCL6-deficient CD4+ T cells as compared to recipients given wild-type CD4+ T cells. These observations are consistent with our notion that extrafollicular CD4+ T and B interaction is critical in induction of cGVHD, and that peripheral Treg cells but not follicular Treg cells can regulate cGVHD development.

The pathogenesis of chronic GVHD is a complex process involving different Th subsets (i.e., Th1, Th2, and Th17), pre-Tfh-like PSGL-1$^{lo}$CD4+ T cells, B cells, and autoantibodies. The current study has revealed the important function of pre-Tfh-like PSGL-1$^{lo}$CD4+ T cells and their interaction with B cells in the pathogenesis of this disease. The mechanisms that enable PSGL-1$^{lo}$CD4+ T cells to cause cGVHD remain unclear. Here we propose a hypothesis to explain the role of pre-Tfh-like PSGL-1$^{lo}$CD4+ T cells in the pathogenesis of cGVHD. As depicted in the diagram (Fig. 9), during cGVHD development, allo- and autoreactive CD4+ T cells first interact with host- or donor-type DCs and differentiate into various Th subsets (i.e., Th1, Th2, and Th17), and some CD4+ T cells differentiate into pre-Tfh cells under the influence of IL-6 and ICOS. The pre-Tfh cells upregulate expression of Stat3, BCL6, IL-21, and CXCR4, but downregulate CCR7 and PSGL-1, and migrate from the T-cell zone to the T–B border. Unlike pre-Tfh cells in the lymphoid tissues of healthy mice that upregulate CXCR5 and migrate into the center of B-cell zone, the pre-Tfh-like PSGL-1$^{lo}$CD4+ T cells in cGVHD recipients maintain low-level expression of CXCR5 and persist at the T–B border, where they interact with donor B cells to produce low-affinity IgG1 with no somatic hypermutation. Pathogenic PSGL-1$^{lo}$CD4+ T cells and deposition of IgG1 antibodies damage lymphoid follicles, thereby preventing GC formation. In addition, pre-Tfh-like PSGL-1$^{lo}$ CD4+ T cells and B cells may migrate into GVHD target tissues, such as skin and lung, to interact and produce IgG antibodies that augment the pathogenesis of cGVHD, which is exacerbated by infiltration of Th1 and Th17 cells[6, 9]. CXCR5$^-$PD-1$^{hi}$ non-Tfh but Tfh-like CD4+ T cells interact with B cells in the inflamed tissues in mice with experimental allergic pneumonia[49] and in humans with rheumatoid arthritis[50]. Further studies are needed to elucidate the surface phenotype and B-cell helper function of PSGL-1$^{lo}$CD4+ T cells in different target tissues and at different stages of cGVHD. In addition, our current studies suggest that blocking the expansion of pre-Tfh-like PSGL-1$^{lo}$CD4+ T cells and augmenting the expansion of Foxp3+ Treg cells by blocking or targeting Stat3 in CD4+ T cells may be an effective approach for preventing or treating cGVHD.

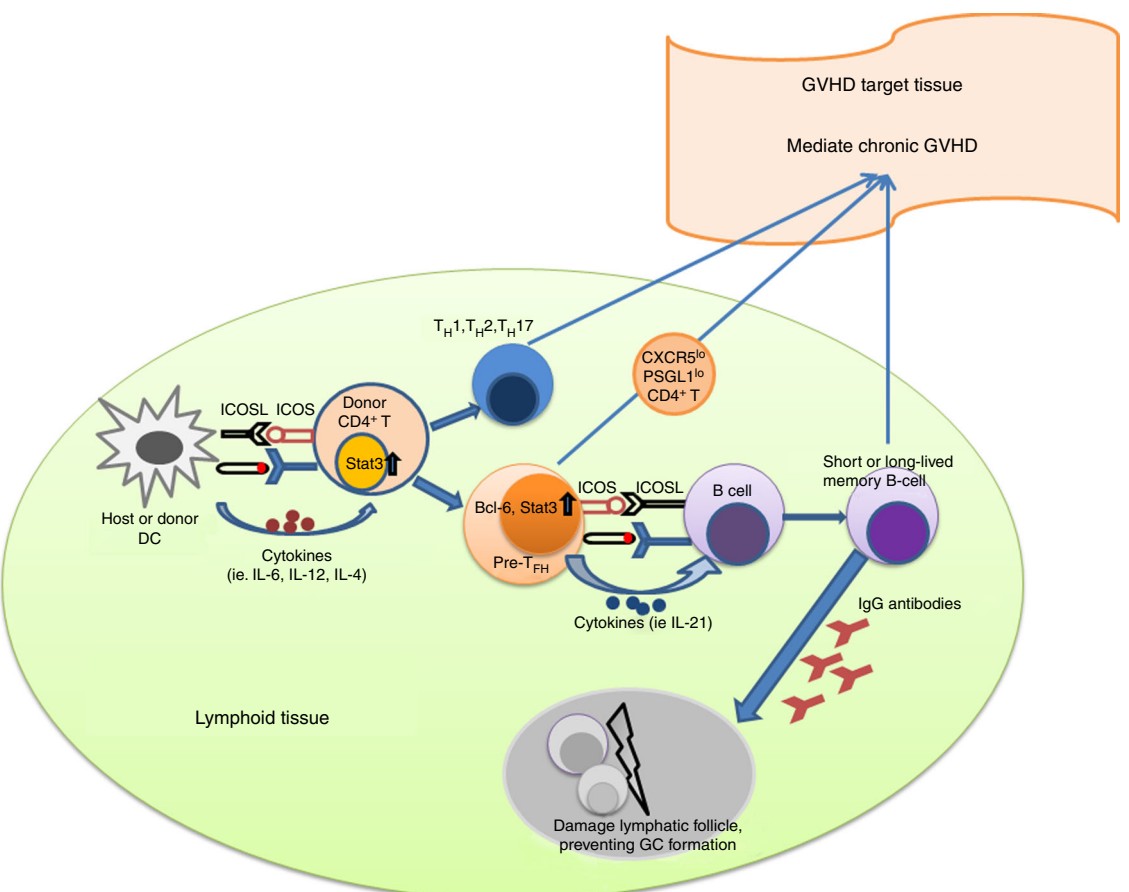

**Fig. 9** Hypothesis on the potential function of PSGL-1$^{lo}$CD4+ T and B interactions in the pathogenesis of chronic GVHD

## Methods

**Mice.** BALB/c (H-2$^d$) and C57BL/6 (H-2$^b$) mice were purchased from National Cancer Institute Laboratories (Frederick, MD). LP/J (H-2$^b$), B10.D2 (H-2$^d$), B10.BR (H-2$^k$) mice were purchased from The Jackson Laboratory (Bar Harbor, ME). BCL6$^{fl/fl}$ Mb1-Cre C57BL/6 were obtained from Dr Markus Muschen's lab at University of California San Francisco (San Francisco, CA)[40]. Stat3$^{fl/fl}$CD4-Cre C57BL/6 were obtained from Dr Hua Yu's lab at City of Hope National Medical Centre (Duarte, CA)[61]. BCL6$^{fl/fl}$ C57BL/6 was mated with CD4-Cre-C57BL/6 to generate BCL6$^{fl/fl}$ CD4-Cre-C57BL/6 mice, and mice over fifth generations were used for experiment. Mice were maintained in a specific pathogen-free room at the City of Hope Research Animal Facilities (Duarte, CA). Eight to 12 weeks old male mice were used for all experiments. Mice were killed using $CO_2$. All animal protocols were approved by the City of Hope Institutional Animal Care and Use Committee.

**Induction and assessment of GVHD.** The mice were irradiated by using a $^{137}$Cs source at a dose of 850 cGy 8–10 h before HCT. Recipients were injected with $2.5 \times 10^6$ T-cell-depleted donor BM cells (TCD-BM) alone or $2.5 \times 10^6$ TCD-BM together with splenocytes $0.01 \times 10^6$ or $1 \times 10^6$ from C57BL/6 donors, 0.25 or $1 \times 10^6$ from B10.BR donors, or $10 \times 10^6$ from LP/J or B10.D2 donors.

The assessment and scoring of clinical cutaneous GVHD were performed according to previous publications with mild modification[7, 9]. Mice were evaluated and scored based on the development of alopecia and ulcers on hair-bearing skin. Ulcers or scaling in non-hair-bearing skin (ears, tails, and paws) were also examined for scoring. (1) skin ulcers with alopecia < 1 cm$^2$ in area; (2) skin ulcers with alopecia 1–2 cm$^2$ in area; (3) skin ulcers with alopecia > 2 cm$^2$; (4) skin ulcers with alopecia > 30% body area.

**Histopathology.** Hematoxylin and eosin (HE) staining on formalin fixed paraffin-embedded tissue slides were used for evaluation. Slides were examined at ×200 magnification, and tissue damage was blindly assessed according to our previous publications[7, 9]. Salivary gland GVHD was evaluated based on mononuclear cell infiltration and structural disruption, with a maximum score of 8. Skin GVHD was scored according to damage in the epidermis and dermis as judged by hyperplasia of epidermis, enlargement, fibrosis of dermis, and loss of subcutaneous fat with the maximum score 9. Lung tissue was evaluated on a scoring system based on perivascular and peribronchiolar infiltration and inflammation; the maximum score is 9. Liver was scored on the number of involved tracts and the severity of lymphocytic infiltration and liver cell necrosis with the maximum score of 9.

**Antibodies for histoimmunofluorescent staining.** FITC-labeled anti-mouse CD3 (145-2C11, #11-0031-82, 1:100) and GL7 (GL7, #13-5902-81, 1:200) were purchased from eBiosciences (San Diego, CA). Rat anti-mouse B220 (RA3-6B2, #550286, 1:100) and rat anti-mouse IgD (11-26c.2a, #553438, 1:100) were purchased from BD Biosciences (San Jose, USA). Biotinylated PNA (#B-1075,1:300), goat anti-mouse IgM (#FI-2020,1:100) and AMCA-labeled streptavidin (#SA-5008,1:300), biotinylated ulex europaeus agglutinin-1 (UEA-1, #B1065, 1:200) and FITC-labeled streptavidin (#SA-5001, 1:100) were purchased from Vector Laboratories (Burlingame, USA). Texas red-labeled goat anti-rat IgG (#T-6392, 1:500), Alexa Fluor® 488-labeled goat anti-rat IgG (#A-11006, 1:500), and Alexa Fluor® 488-labeled goat anti-mouse IgG (#A11001, 1:200) were purchased from Thermo Fisher Scientific (Waltham, USA). Rat anti-mouse cytokeratin 8 (#Troma-1 1:100) was purchased from Developmental Studies Hybridoma Bank (Iowa City, USA).

**GC detection.** Spleen was embedded in optimum cutting temperature compound and 5 μm thickness cryosections were used for staining. Three staining combinations were performed to detect GCs according to previous publications in order to ensure the reliability of staining. Combination 1: cryosections were stained with FITC-labeled anti-mouse CD3 combined with rat anti-mouse B220 and biotinylated PNA, followed by Texas red-labeled goat anti-rat IgG secondary antibody and AMCA-labeled streptavidin. Combination 2: cryosections were stained with FITC-labeled goat anti-mouse IgM and biotinylated PNA followed by AMCA-streptavidin. Combination 3: cryosections were stained with rat anti-mouse IgD and biotinylated GL7 followed with Alexa Fluor® 488-labeled goat anti-rat IgG secondary antibody and AMCA-streptavidin. Images were acquired on Olympus BX50 immunofluorescent microscope (Olympus, Center Valley, USA) at ×200 magnification.

**IgG deposition and thymic epithelial cells staining.** The detection of IgG deposition in skin and thymus tissues, and thymus epithelial cell staining was performed according to previous publications of our group and others with modest modification[9, 43]. Skin and thymus cryosections were stained with Alexa Fluor® 488-labeled goat anti-mouse IgG for IgG deposition. Thymus cryosections were stained with biotinylated ulex europaeus agglutinin-1 (UEA-1) for medullary epithelial cells, and rat anti-mouse cytokeratin 8 for cortical epithelial cells followed by FITC-streptavidin or Texas red-labeled goat anti-rat IgG. Images were acquired on Olympus BX50 immunofluorescent microscope (Olympus, Center Valley, USA) at ×200 magnification.

**Real-time quantitative PCR.** RNA samples were isolated from sorted CD4$^+$CD44$^{hi}$PSGL-1$^{lo}$CD62L$^{lo}$ T cells by using miRNeasy Mini Kit purchased from Qiagen (Valencia, CA, #217004). Real-time qualitative PCR were performed using SYBR Green Supermix (Bio-Rad, Hercules, CA #172-5124) after RNA was reverse transcribed into cDNA with reverse transcription kits (Bio-Rad, Hercules, CA, #170-8890). Relative gene expression was normalized to GAPDH. The primers used for CCR7, CXCR4, CXCR5, BCL6, STAT3, ICOS, PDCD-1,CD80, PD-L1, and GAPDH amplification have been described in previous publications by our group and others[62–69].

The detailed sequence was listed as below:
CCR7:
Forward 5′-TGGTGGCTCTCCTTGTCATTT-3′,
Reverse 5′-ACCGACTCGTACAGGGTGTAGTC-3′.
CXCR4:
Forward 5′-CATGGAACCGATCAGTGTGAGT-3′,
Reverse 5′-GCAGGGTTCCTTGTTGGAGT-3′.
CXCR5:
Forward 5′-ACTCCTTACCACAGTGCACCTT-3′,
Reverse 5′-GGAAACGGGAGGTGAACCA-3′.
BCL6:
Forward 5′-CACACCCGTCCATCATTGAA-3′,
Reverse 5′-TGTCCTCACGGTGCCTTTTT-3′.
STAT3:
Forward 5′-ACCAACATCCTGGTGTCTCC-3′,
Reverse 5′ TTATTTCCAAACTGCATCAATGA-3′.
ICOS:
Forward 5′-CTCACCAAGACCAAGGGAAGC-3′,
Reverse 5′-CCACAACGAAAGCTGCACACC-3′.
PDCD-1:
Forward 5′-CGTCCCTCAGTCAAGAGGAG-3′,
Reverse 5′-GTCCCTAGAAGTGCCCAACA-3′.
CD80:
Forward 5′GGCAAGGCAGCAATACCTTA 3′
Reverse 5′ CTCTTTGTGCTGCTGATTCG 3′;
PD-L1:
Forward 5′-AGGATATTTGCTGGCATTATATTCAC-3′,
Reverse 5′-ACAAACTGAATCACTTGCTCATCTT-3′
GAPDH:
Forward 5-TCACCACCATGGAGAAGGC-3,
Reverse 5-GCTAAGCAGTTGGTGGTGCA-3.

**RNA sequencing analysis.** RNA samples were isolated from sorted CD4$^+$CD44$^{hi}$PSGL-1$^{lo}$CD62L$^{lo}$ T cells by using miRNeasy Mini Kit purchased from Qiagen (Valencia, USA, #217004). Total RNA sequencing was performed and analyzed by the Integrative Genomics Core, City of Hope National Medical Center (Duarte, CA). Transcriptome libraries were constructed with TruSeq Stranded Total RNA Ribo-Zero Kit (Illumina, CA, #RS-122-2203). In brief, 500 ng of total RNA from each sample was used to construct a cDNA library, followed by sequencing on the Illumina Hiseq 2500 with single end 50 bp reads according to the manufacturer's recommendations. The sequences were aligned to mouse genome assembly mm9 using Tophat2. For each sample, total counts for Refseq genes were summarized by HTseq[70] and reads per kilobase of transcript per million mapped reads (RPKM) were calculated. The RNA expression levels of the genes of interest were shown as mean centered log$_2$ RPKM by heat maps generated by Java Treeview. Counts were normalized, and differential expression analysis between groups was conducted by using the Bioconductor package "edgeR".

**Flow cytometry analysis and cell sorting.** The antibodies and reagents used for flow cytometry analysis and cell sorting are listed as below: antibodies to mouse, CD4 (RM4-5, #48-0042-82, 1:100), CD44 (IM7, #25-0441-81, 1:300), CD62L (MEL-14, #11-0621-85, 1:100), CD19 (eBio 1D3, #11-0193-81, 1:100), GL7 (GL7, #13-5902-81, 1:200), PD-1 (J43, #17-9985-82, 1:200), CD23 (B3B4, #13-0232-81, 1:200), IgM (II/41, #17-5790-82, 1:200), IgD (11-26c, #12-5993-82,1:300), interleukin 21 (FFA1, #12-7211-80,1:100), interleukin-13 (eBio13A, #12-7133-82, 1:100), interferon-γ (XMG1.2, #12-7311-82, 1:100), interleukin-17 (eBio 17B7, #12-7177-81, 1:100), ICOS (CD278, 7E.17G9, #12-9942-81, 1:200), BCL-6 (BCL-DWN, #12-5453-80, 1:100), FasL (CD178, MFL3, #13-5911-82, 1:100), and rat IgG 2a isotype control antibody (eBR2a, #12-4321-81, 1:100), anti-rat IgG2b (R2B-7C3, #13-4815-80, 1:100), streptavidin (#13-4317-82, 1:300) were all purchased from eBiosciences (San Diego, CA). Anti-mouse ICOSL (CD275, HK5.3, #107403, 1:100) was purchased from Biolegend (San Diego, USA). Fas (Jo2, #554258, 1:200), CD21 (7G6, #561769, 1:100), PSGL-1 (2PH1, #562806 1:300) and CXCR5 (2G8, #551960, 1:200) were purchased from BD Biosciences (San Jose, USA). Antibody to mouse CCR9 (242503, #FAB2160P, 10 μl/10$^6$ cell) was purchased from R&D (Minneapolis, USA). FoxP3 staining kit (00-5523-00) was purchased from eBioscience (San Diego, CA). Aqua fluorescent reactive dye for viability analysis (L34957) was obtained from Invitrogen (Carlslad, CA). All staining was performed according to the manufacturer's instructions. Flow cytometric analysis was performed on CyAn Immunocytometry system (DAKO Cytomation, Fort Collins, CO) and data were analyzed with FlowJo software (Tree Star, Ashland, OR). Extrafollicular PSGL-

$1^{lo}CD4^+$ cells were sorted with a 6-laser Arial III immunocytometry system (DakoCytomation, Fort Collins, USA) and used for intracellular cytokine staining, RNA-seq analysis and in vivo adoptive transfer experiments. The gating strategies were shown in Supplementary Fig. 25.

**In vivo anti-ICOS treatment**. Anti-mouse ICOS (7E.17G9) was purchased from Bio-X cell (West Lebanon, USA, #BE0059). The in vivo anti-ICOS treatment was performed according to previous report with modification[28]. The treatment started on the day of transplantation, and recipients were injected i.p. with 200 μg/mouse anti-mouse ICOS or isotype control (rat IgG2b, #BE0090) every other day until 45 days after transplantation.

**Statistical analysis**. Survival in different groups was compared using the log-rank test with the program GraphPad Prism, version 6.0 (GraphPad Software, San Diego, CA). Two-way ANOVA was used for comparing body weight changes and cutaneous cGVHD score in different groups. Unpaired two-tailed Student's $t$ test was used to determine significant differences between two experimental groups.

**Data availability**. The total RNA sequencing data have been deposited and are available in the GEO database (GSE101552).

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

## Acknowledgements

We thank Dr Arthur Riggs for providing additional fund to support this project. We thank Lucy Brown and her staff at the COH Flow Cytometry Facility, Sofia Loera and her staff at COH Anatomic Pathology Core, and Dr Richard Ermel and his staff at COH Animal Research Center for providing excellence service. This work was supported by National Institutes of Health Grant R01-AI066008 and R56-AI066008 (to D.Z.) and supported by the National Cancer Institute of the National Institutes of Health under award number P30CA033572.

## Author contributions

R.D. designed and performed research as well as prepared the manuscript; C.H. and G.X. assistance on generation of B-BCL6KO mice; Q.S. assistance on experiments; C.Y. assistance on generation of CD4-Stat3 KO mice; H.Y. advice on using CD4-Stat3 KO mice and critical review of manuscript; M.M. advice on using B-BCL6KO and CD4-BCL6KO mice and critical review of manuscript. X.W. advice on RNA-seq experimental design and data analysis and interpretation; S.F. advice on the project and critical review of manuscript; P.M. advice on experimental design, critical review, and editing of manuscript. D.Z. designed and supervised the research and wrote the manuscript.

## Additional information

**Competing interests:** The authors declare no competing financial interests.

