## [Peer Review File · Nature Communications]

Reviewers' comments:

Reviewer #1 (aGVHD, DAMP/inflammasome) (Remarks to the Author):

Deng et al. report on the influence of BCL6 and STAT3 deficiency in CD4 T cells on germinal center formation and occurrence of graft versus host disease. The manuscript is well written but has several shortcomings in particular with respect to novelty and methods.

1. A major weakness of the paper is the novelty. Multiple groups have already shown that blocking ICOS or using ICOS deficient donors reduces acute and chronic GVHD. Effector T cells in general show reduced proliferation and activation upon anti-ICOS treatment.

Flynn R. et al. Increased T follicular helper cells and germinal center B cells are required for cGVHD and bronchiolitis obliterans. *Blood* 123, 3988–3998 (2014).

Taylor PA, Friedman TM, Korngold R, Noelle RJ, Blazar BR.. Tolerance induction of alloreactive T cells via ex vivo blockade of the CD40:CD40L costimulatory pathway results in the generation of a potent immune regulatory cell. *Blood* (2002) 99:4601–9.10.1182/blood.V99.12.4601

Li J, Semple K, Suh WK, et al. Roles of CD28, CTLA4, and inducible costimulator in acute graft-versus-host disease in mice. *Biol Blood Marrow Transplant.* 2011;17:962

Fujimura J, Takeda K, Kaduka Y, et al. Contribution of B7RP-1/ICOS co-stimulation to lethal acute GVHD. *Pediatr Transplant.* 2010;14:540

Mollweide A, Staeger MS, Hoeschen C, Hideo Y, Burdach S, Richter GH. Only therapeutic ICOS:ICOSL blockade alleviates acute graft versus host disease. *Klin Padiatr.* 2009;221:344

2. Another weakness is the use of a well-known acute GVHD model (B6 into BALB/c) while the authors state that they investigate chronic GVHD. They considered the skin inflammation as a typical sign for chronic GVHD but acute GVHD is affecting the skin as well. Only in some initial experiment the authors have used a chronic GVHD model (LP/J \times C57BL/6).

A previous study by the same authors used the same model but named it acute GVHD model. The survival of the WT is very similar to the model that is described in the manuscript as cGVHD.

Yi T1, Zhao D, Lin CL, Zhang C, Chen Y, Todorov I, LeBon T, Kandeel F, Forman S, Zeng D. Absence of donor Th17 leads to augmented Th1 differentiation and exacerbated acute graft-versus-host disease. *Blood.* 2008 Sep 1;112(5):2101-10

2. Typical signs of cGVHD are not studied such as antibody deposition, Th17, sclerodermatitis, macrophage infiltration, fibrosis or other.

3. Figure 6: Splenocytes from either WT or B-BCL-6-/- C57BL/6 donors are used - therefore the authors cannot distinguish between effects caused by the BCL-6 deficient T cells or B-cells or other splenic immune cells of donor origin.

4. The authors observed the absence of GC in mice developing GVHD. In previous studies of chronic GvHD Flynn et al and Srinivasan et al. showed that GCs are still there in chronic GVHD and are required for disease pathology. The authors only briefly mentioned the studies. What is different between these studies? Can you exclude that your results are due to acute GvHD?

Flynn R. et al. Increased T follicular helper cells and germinal center B cells are required for cGVHD and bronchiolitis obliterans. *Blood* 123, 3988–3998 (2014).

Srinivasan M, Flynn R, Price A, et al. Donor B-cell alloantibody deposition and germinal center formation are required for the development of murine chronic GVHD and bronchiolitis obliterans. *Blood* 2012;119(6):1570-1580

5. The adequate control for expansion of PSGL-1^{lo}CD4⁺ cells would be a syngeneic transplantation. The additional transplantation of 1x10⁶ splenocytes in GVHD recipients in comparison to no-GVHD recipients raises the question if the increased number that was observed is just by homeostatic proliferation and survival of these cells or by alloantigen based activation. Syngeneic transplantation of 1x10⁶ splenocytes would also account for the additional transplanted T cell numbers and the homeostatic proliferation.

6. The microarray data are not very convincing with n=2. They have to be confirmed by quantitative RT-PCR using independent samples and on the protein level.

7. Statistical tests for GVHD score, Body weight and survivals are missing.

Minor comments:

...pathogenic CD4⁺ T and B cells, but the role of extrafollicular...

Chronic GVHD often follows acute GVHD. Stimmt das?

...cGVHD recipients... cGVHD patients

Introduction to much/complicated/detailed

BALB/c recipients were injected with TCD-BM plus 1x10⁶ or 0.01 x10⁶ C57BL/6 spleen cells.

Reviewer #2 (Tfh, Bcl6) (Remarks to the Author):

This study from Deng et al provides some novel information about the role of TFH-like cells in chronic graft versus host disease (cGVHD), using a mouse model system. The major claims of the paper are showing that extra-follicular CD4 T cells are involved in cGVHD pathogenesis and that cGVHD can be blocked by inhibiting these cells via an ICOS blockade, or via genetic disruption of the transcription factors Bcl6 or Stat3 in T cells. These findings are important for the understanding of cGVHD and also help to clarify respective roles of germinal center-associated TFH cells and extra-follicular TFH-like cells. However, there are a number of problems in the manuscript that weaken these conclusions, and a number of other concerns.

The following issues should be addressed in a revision:

- 1) What is the relationship of TFH cells and extra-follicular TFH-like cells in this model? In every mouse experiment, there should be a direct quantitation of TFH cells (CXCR5⁺ CD62L^{low}, PSGL-1^{low}) and extra-follicular TFH-like cells (CXCR5⁺CD62L^{low}, PSGL-1^{low}). It also would be useful to compare the expression of PD-1 and ICOS on these two populations.
- 2) What is the cytokine profile of the extra-follicular TFH-like cells in the cGVHD system? Does the cytokine profile change over the course of the disease?
- 3) Are the extra-follicular TFH-like cells promoting pathology via helping B cells make auto-Ab or are they making inflammatory cytokines that drive disease independent of Ab?
- 4) The nomenclature used for the "no GVHD" groups are confusing in Figures 2, 6, 7. What are in the groups that say, for instance, "B-BCL-6^{+/+} -noGVHD"? What is the difference between the "B-BCL-6^{+/+}-noGVHD" and "B-BCL-6^{-/-} -noGVHD" groups?
- 5) Figure 3C purports to show a difference in CXCR4 expression but both groups just have black boxes with no apparent differences.
- 6) Figure 7A is missing a label
- 7) When are spleens analyzed in Figure 7G?
- 8) In figure 8, the cytokine arrow between pre-TFH and B cells should be going from the pre-TFH to B

cells, not other way as shown

9) It should be noted that deletion of Bcl6 and Stat3 with CD4-cre affects Tregs, and specifically inhibit TFR cell generation.

10) References should be given for the source of the Bcl6-flox mice and the Stat3-flox mice

11) In discussion "T-B boarder" is written twice. It should be "T-B border"

Reviewer #3 (ICOS, cytokine signaling, Tfh) (Remarks to the Author):

T follicular helper (Tfh) cells have emerged as a distinct subset/lineage of CD4+ T cells that are primarily responsible for providing help to B cells during immune responses to TD Ag and mediating the differentiation of B cells into memory and plasma cells during GC reactions in secondary lymphoid tissues. The critical function of Tfh cells is evident from animal models and human diseases where Tfh cells are reduced/absent and humoral immunity is impaired, while on the other hand excessive numbers/production or function of Tfh cells has been associated with, and likely causes, autoimmune conditions such as SLE etc. however, there are other "types" of CD4 T cells that also provide help to B cells that are not strictly located within the B-cell follicles – these have been coined extrafollicular CD4+ T cells, and were first identified and characterized by Joe Craft in the setting of a murine model of autoimmunity. Since then, other groups have identified analogous cells in normal humoral immune responses, implicating these cells in the early stages of B-cell activation and differentiation (pre-germinal center).

In this current study, the authors have used numerous models of graft vs host disease and detailed physiological, developmental, cellular and functional analyses to reveal a critical role for extrafollicular CD4 T cells in disease pathogenesis. Importantly (but perhaps predictably) the generation of pathogenic extrafollicular CD4 T cells could be reduced by blocking ICOS/ICOS-L interactions, as well as genetic ablation of Bcl-6 or STAT3 in CD4 T cells. Overall, this study sheds substantial light on etiology of chronic GVHD, at least in murine models – it will need to be determined whether these findings are directly relevant to human GVHD. This notwithstanding, the study is well and comprehensively performed, and contains some important novel findings. Several comments follow that should be used as a guide to improve the novelty, mechanisms and conclusions of the findings.

1. The finding relating to the effects of STAT3 deficiency on disease need to be extended. First, what was the level of expression of Bcl-6 in Stat3-deficient CD4 T cells (and extrafollicular CD4 T cells specifically)? Second, Stat3 deficiency did not overcome acute GVHD, but greatly improved chronic GVHD. This raises questions about the nature and quality of extrafollicular CD4 T cells during these distinct phases of disease progression. Are these cells less likely to be pathogenic in the early/acute phase of GVHD? Third, Stat3 deficiency not only reduced pathogenic extrafollicular CD4 T cells but also increased Tregs. However, these latter cells were only assessed by phenotype – not function. Were the Stat3-deficient Tregs superior to WT Tregs at suppressing T cell activation and function? Was the increase in Tregs protective ie did disease severity worsen if Tregs were depleted from mice harbouring Stat3-deficient CD4 T cells?

2. several of the Figures contain results from data for gene expression from RNA Seq. these data need to be confirmed by Western blot or FACS. Eg Fig 3C – for CXCR4, CXCR5 and CCR7; Fig 4A – for many of the surface receptors listed; Fig 7 for STAT3 and BCL6. Also for Fig 7A, it is not indicated which panels correspond to which experimental mice.

3. for the expts for Fig 5D, it is possible that the reduced level of detected ICOS expressed by the labelled cells by FACS in the mice receiving the anti-ICOS mAb reflects masking by the blocking ICOS mAb that was still bound to the cells. this needs to be determined. Also for this data the treatment

regime was extreme – blocking ICOS Ab injected every 2nd day for 45 days. Was this necessary? Was there a shorter time frame in which the ICOS/ICOS-L interaction could be blocked and a physiological effect still be detected?

REVIEWERS' COMMENTS:

Reviewer #1 (Remarks to the Author):

The authors have clarified some issues.

Reviewer #2 (Remarks to the Author):

As I wrote previously, the major claims of the paper are showing that extra-follicular CD4 T cells are involved in cGVHD pathogenesis and that cGVHD can be blocked by inhibiting these cells via an ICOS blockade, or via genetic disruption of the transcription factors Bcl6 or Stat3 in T cells. These findings are important for the understanding of cGVHD and also help to clarify respective roles of germinal center-associated TFH cells and extra-follicular TFH-like cells.

The revised manuscript addresses my most pressing concerns, but I still find it disconcerting that the authors do not have flow cytometry comparing Tfh versus the supposed extra-follicular Tfh-like cells. However, the RNA expression data does address CXCR5 expression and basically addresses the issue.

The legends are better, but in Figure 1, it should be laid out clearly what Group 1 is versus Group 2 versus Group 3. Also, TCD should be defined somewhere in the main manuscript.

In summary, the main conclusions of the paper are significant for the field, and convincingly presented. Statistics seem fine and there is probably enough detail given for reproduction by other researchers.

Reviewer #3 (Remarks to the Author):

The authors have adequately addressed most of the concerns raised in the original review of their manuscript. However, they did not assess/confirm differential gene expression by FACS, as requested - rather they have confirmed these differences by qPCR. This should suffice - though it would have been preferable to see either protein expression (ie FACS analyses) or simply for the authors to state that they felt the data presented as qPCR was suitable.

REVIEWERS' COMMENTS:

Reviewer #1 (aGVHD, DAMP/inflammasome) (Remarks to the Author):

Deng et al. report on the influence of BCL6 and STAT3 deficiency in CD4 T cells on germinal center formation and occurrence of graft versus host disease. The manuscript is well written but has several shortcomings in particular with respect to novelty and methods.

1. A major weakness of the paper is the novelty. Multiple groups have already shown that blocking ICOS or using ICOS deficient donors reduces acute and chronic GVHD. Effector T cells in general show reduced proliferation and activation upon anti-ICOS treatment.

Flynn R. et al. Increased T follicular helper cells and germinal center B cells are required for cGVHD and bronchiolitis obliterans. *Blood* 123, 3988–3998 (2014).

Taylor PA, Friedman TM, Korngold R, Noelle RJ, Blazar BR.. Tolerance induction of alloreactive T cells via ex vivo blockade of the CD40:CD40L costimulatory pathway results in the generation of a potent immune regulatory cell. *Blood* (2002) 99:4601–9.10.1182/blood.V99.12.4601

Li J, Semple K, Suh WK, et al. Roles of CD28, CTLA4, and inducible costimulator in acute graft-versus-host disease in mice. *Biol Blood Marrow Transplant.* 2011;17:962

Fujimura J, Takeda K, Kaduka Y, et al. Contribution of B7RP-1/ICOS co-stimulation to lethal acute GVHD. *Pediatr Transplant.* 2010;14:540

Mollweide A, Staeger MS, Hoeschen C, Hideo Y, Burdach S, Richter GH. Only therapeutic ICOS:ICOSL blockade alleviates acute graft versus host disease. *Klin Padiatr.* 2009; 221:344

We regret that we have created a misunderstanding about the main focus of our manuscript.

We agree that it has been reported by Flynn and Blazar et al that there were enlarged germinal center in the lymphoid tissues of chronic GVHD recipients at the disease onset, and those publications are cited in the introduction and discussion (see pages 4 and 16). However, in our report, we have clearly demonstrated that there is no enlargement of germinal centers in the chronic GVHD recipients. Instead, lymphofollicles and germinal centers were destroyed in chronic GVHD recipients at the disease onset. This discrepancy is not due to model difference, because we have checked all chronic GVHD models reported by Flynn et al, and we have consistently found destruction of lymphofollicles and loss of germinal centers (see Supplemental Figures 4,5, 6 and 7). Furthermore, by using donor transplants with specific BCL6 deficiency in B cells that is not able to give rise to germinal centers, we have demonstrated that germinal center formation is dispensable for development of chronic GVHD (Fig 2).

We also agree that others have shown amelioration of acute and chronic GVHD by blockade of ICOS interaction with its ligand, and these publications have been incorporated into our text (see page 17). However, by using B-BCL6 deficient transplants that cannot give rise to germinal centers, we have specifically demonstrated that extrafollicular CD4⁺ T and B cell interactions also depend on ICOS/ICOS ligand interactions. Therefore, blockade ICOS/ICOS ligand interaction amelioration of chronic GVHD in the report by Flynn et al may not result from prevention of germinal center formation, because our studies showed that there was no germinal center formation in those models. Instead, prevention of chronic GVHD may result from blocking extrafollicular CD4⁺ T and B interaction. This point has been incorporated into discussion (see page 17).

Taken collectively, the novelty of our report is very high, because it has clarified a highly controversial question about the role of germinal center formation in chronic GVHD pathogenesis. Our report has highlighted novel mechanisms for the pathogenesis of autoimmune-like chronic GVHD that are likely to be relevant to other autoimmune conditions.

2. Another weakness is the use of a well-known acute GVHD model (B6 into BALB/c) while the authors state that they investigate chronic GVHD. They considered the skin inflammation as a typical sign for chronic GVHD but acute GVHD is affecting the skin as well. Only in some initial experiment the authors have used a chronic GVHD model (LP/J to C57BL/6). A previous study by the same authors used the same model but named it acute GVHD model. The survival of the WT is very similar to the model that is described in the manuscript as cGVHD.

Yi T1, Zhao D, Lin CL, Zhang C, Chen Y, Todorov I, LeBon T, Kandeel F, Forman S, Zeng D. Absence of donor Th17 leads to augmented Th1 differentiation and exacerbated acute graft-versus-host disease. *Blood*. 2008 Sep 1;112(5):2101-10

We regret the misunderstanding about our chronic GVHD model.

*We agree with the reviewer that the HCT model of C57BL/6 donor to BALB/c recipient has been traditionally considered as an acute GVHD model by us and by other investigators. However, in a previous report (Wu et al, *J. Immunol* 2013), we have demonstrated that after titrating down the numbers of C57BL/6 donor spleen cells or T cells in the graft, BALB/c recipients survived beyond the acute GVHD phase and emerged into a well characterized chronic GVHD phase. The recipients survived for more than 60 days after transplantation with grafts containing low numbers of donor spleen or T cells, and they developed clinical signs and histopathology that reflect manifestations of human chronic GVHD, including scleroderma, lymphocytic bronchitis, and damage in the salivary and lacrimal glands. In addition, we showed with other models that induction of chronic GVHD does not depend on the donor and recipient strain combination, but it does depend on survival time after HCT. This new concept about induction of chronic GVHD in murine models has been recognized by many investigators and has been cited by a lot of publications including a recent review (Socié G, Ritz J. Current issues in chronic graft-versus-host disease. *Blood* 2014;124(3):374-384). We have further clarified our murine models of chronic GVHD in the revised manuscript (see page 5)*

2. Typical signs of cGVHD are not studied such as antibody deposition, Th17, sclerodermatitis, macrophage infiltration, fibrosis or other.

We understand that antibody deposition, Th17, scleroderma, macrophage infiltration and fibrosis are involved in chronic GVHD pathogenesis. However, those factors are not the focus of current studies. The focus of our studies are on using well described chronic GVHD models to unravel the role of follicular and extrafollicular donor CD4⁺ T and B interactions in the pathogenesis of chronic GVHD. Therefore, we include those factors only when it is necessary. For example, we compared antibody deposition in the thymus and skin tissues to help reveal the role of specific Stat-3 deficiency in donor CD4⁺ T cells in prevention of chronic GVHD (see newly added data Fig. S23 and page14, first paragraph).

3. Figure 6: Splenocytes from either WT or B-BCL6^{-/-} C57BL/6 donors are used - therefore the authors cannot distinguish between effects caused by the BCL-6 deficient T cells or B-cells or other splenic immune cells of donor origin.

We regret the misunderstanding here. We would like to clarify that comparison of WT and specific BCL6 deficiency in B cells (B-BCL6^{-/-}) is with Figure 2. In Figure 6, we used spleen cells from WT donors or from

donors with BCL6 deficiency specifically in CD4⁺ T cells, because we intended to study the role of BCL6 in donor CD4⁺ T cells in regulating follicular and extrafollicular CD4⁺ T and B cell interactions. Thus, it is appropriate to use whole spleen cells with specific BCL-6 deficiency in CD4⁺ T cells.

4. The authors observed the absence of GC in mice developing GVHD. In previous studies of chronic GvHD Flynn et al and Srinivasan et al. showed that GCs are still there in chronic GVHD and are required for disease pathology. The authors only briefly mentioned the studies. What is different between these studies? Can you exclude that your results are due to acute GvHD?

Flynn R. et al. Increased T follicular helper cells and germinal center B cells are required for cGVHD and bronchiolitis obliterans. Blood 123, 3988–3998 (2014).

Srinivasan M, Flynn R, Price A, et al. Donor B-cell alloantibody deposition and germinal center formation are required for the development of murine chronic GVHD and bronchiolitis obliterans. Blood 2012;119(6):1570-1580

We have measured germinal center formation at chronic GVHD phase, day 60 after HCT, as described by Flynn et al or Srinivasan et al. We used the same models of donor to recipient and same staining method, as well as additional models, but as shown in main figures 1 as well as supplementary figures 2, 3, 4, 5, 6 and 7, we were unable to replicate results reported by Flynn et al. and Srinivasan et al. We have never found enlarged germinal centers in chronic GVHD recipients, as compared to control no-GVHD recipients given TCD-BM cells only. We observed much smaller germinal centers in recipients with very mild chronic GVHD. What's more chronic GVHD was successfully induced in recipients which lack of germinal center formation (Figure 2) Therefore, our data indisputably showed that chronic GVHD is associated with destruction or loss of germinal centers and was not associated with enlarged germinal centers (see result section page 6-7 and discuss section page 16).

We agree that mild or non-lethal acute GVHD is likely to damage the lymphoid structure and prevent germinal center formation in chronic GVHD recipients (see discussion section page 17, first paragraph).

5. The adequate control for expansion of PSGL-1^{lo}CD4⁺ cells would be a syngeneic transplantation. The additional transplantation of 1x10⁶ splenocytes in GVHD recipients in comparison to no-GVHD recipients raises the question if the increased number that was observed is just by homeostatic proliferation and survival of these cells or by alloantigen based activation. Syngeneic transplantation of 1x10⁶ splenocytes would also account for the additional transplanted T cell numbers and the homeostatic proliferation.

We found that there was no expansion of PSGL-1^{lo}CD4⁺ T cells in syngeneic transplantation recipients (see newly added data Figure S14 and the Results on page 10, second paragraph).

6. The microarray data are not very convincing with n=2. They have to be confirmed by quantitative RT-PCR using independent samples and on the protein level.

Each sample represented the sorted PSGL-1^{low}CD4 T cells pulled from 8 mice, and two replicate experiments are shown. The related RNA-seq data are validated with real time PCR (see supplemental Fig. S12, S15 and S18)

7. Statistical tests for GVHD score, Body weight and survivals are missing.

Statistical analysis for GVHD score, bodyweight and survival have been added and indicated in corresponding figure legends (see Figures 1,4,6,7 and pages 31,34,36,37).

Minor comments:

...pathogenic CD4+ T and B cells, but the role of extrafollicular...
Chronic GVHD often follows acute GVHD. Stimmt das?

Yes, it is true based on our best knowledge.

...cGVHD recipients... cGVHD patients

We did not understand the question and cannot address this point.

Introduction to much/complicated/detailed.

We intend to prepare sufficient background information for broad readership to help their understanding of this report, since Nature Communications has broader readership as compared to Blood.

BALB/c recipients were injected with TCD-BM plus 1×10^6 or 0.01×10^6 C57BL/6 spleen cells.

Yes, it is true!

Reviewer #2 (Tfh, Bcl6) (Remarks to the Author):

This study from Deng et al provides some novel information about the role of TFH-like cells in chronic graft versus host disease (cGVHD), using a mouse model system. The major claims of the paper are showing that extra-follicular CD4 T cells are involved in cGVHD pathogenesis and that cGVHD can be blocked by inhibiting these cells via an ICOS blockade, or via genetic disruption of the transcription factors Bcl6 or Stat3 in T cells. These findings are important for the understanding of cGVHD and also help to clarify respective roles of germinal center-associated TFH cells and extra-follicular TFH-like cells. However, there are a number of problems in the manuscript that weaken these conclusions, and a number of other concerns.

The following issues should be addressed in a revision:

1) What is the relationship of TFH cells and extra-follicular TFH-like cells in this model? In every mouse experiment, there should be a direct quantitation of TFH cells (CXCR5+ CD62L-low, PSGL-1-low) and extra-follicular TFH-like cells (CXCR5-CD62L-low, PSGL-1-low). It also would be useful to compare the expression of PD-1 and ICOS on these two populations.

We greatly appreciate the very insightful comment. The relationship of TFH and extra-follicular TFH-like cells in chronic GVHD mice remains unclear, although we have uncovered some hints. Further studies in the future are required. As depicted in summary diagram (Fig. 9), we speculate that extra-follicular TFH-like cells in autoimmune-like chronic GVHD mice are "Pre-TFH-like cells". Due to damage of lymphoid niches by alloreactive and autoreactive T cells, Pre-TFH and B interaction in lymphoid tissues may not lead to formation of germinal centers in lymphoid tissues. Instead, the Pre-TFH cells become $CD44^{hi}CD62L^{lo}PSGL-1^{lo}$ cells and infiltrate GVHD target tissues, where they continue to interact with B cells. This hypothesis is supported by our observations: 1) We found very little germinal center formation in chronic GVHD mice at disease onset. 2) The percentage of $PSGL-1^{lo}CD4^{+}$ T cells peaked in the spleen ~21 days after HCT and peaked in GVHD target tissues (i.e. lung and liver) at ~30 days after HCT when chronic GVHD manifestations first appear. 3) Differentiation of $PSGL-1^{lo}CD4^{+}$ T cells is Stat3 and BCL6-dependent. Recipients given transplants from donors with specific BCL6 deficiency in $CD4^{+}$ T cells showed little cutaneous cGVHD, but add-back of $PSGL-1^{lo}CD4^{+}$ T cells from recipients that cannot give rise to TFH cells restored cutaneous chronic GVHD (newly added data, Fig. S17 in result section).

In a murine model of allergic pneumonia, classical CXCR5⁺BCL6⁺ or CXCR5⁺PD-1⁺ TFH cells were found in the draining LN, but not found in the inflamed lung tissue. However, CXCR5⁺PD-1⁺ICOS⁺CD40L⁺ TFH-like CD4⁺ T cells were identified in the lung tissue, and these cells can help B cells to produce IgG1 and IgA (Van et al: Nature Communications 2016). Additionally, CXCR5⁺PD-1^{hi}CD4⁺ T cells from arthritis synovium of patients with rheumatoid arthritis promote plasma cell differentiation through IL-21 and SLAMF5 interactions (Rao et al: Nature 2017). We would anticipate considerable difficulty in attempting to conduct similar studies in our model, due to the lack of CXCR5⁺PD-1⁺ TFH cells in both lymphoid and GVHD target tissues of mice with chronic GVHD. As an alternative for our future studies, we plan to compare the surface phenotype and function of PSGL1^{lo} and PSGL1^{hi} CD4⁺ T cells in helping B cells in lymphoid tissues and GVHD target tissues of chronic GVHD recipients.

These points have been incorporated into the Discussion (See pages 18).

We appreciate reviewer's suggestion to do a direct quantitation of TFH cells (CXCR5⁺CD62L⁺PSGL-1^{lo}) and extra-follicular TFH-like cells (CXCR5⁺CD62L⁺PSGL-1^{lo}) in each experiment and to measure PD-1 and ICOS expression by the two populations. These are helpful suggestions for future experiments. It would not be feasible for us to redo all of the experiments with these measures for this revision. In addition, doing those experiments is not critical to support the major conclusions of the current manuscript, namely 1) whether germinal center formation is required for induction of chronic GVHD; and 2) whether extrafollicular PSGL-1^{lo}CD4⁺ T cells play a critical role in induction of chronic GVHD. To further strengthen our major conclusion, we have performed add-back experiments. Injection of PSGL-1^{lo}CD4⁺ T cells from chronic GVHD recipients to recipients lacking PSGL-1^{lo}CD4⁺ T cells and cutaneous chronic GVHD. We found that injection of PSGL-1^{lo}CD4⁺ T cells restored cutaneous chronic GVHD (see Supplemental figure S17 on page 13 in result section).

2) What is the cytokine profile of the extra-follicular TFH-like cells in the cGVHD system? Does the cytokine profile change over the course of the disease?

We used intracellular cytokine flow cytometry and found that PSGL-1^{lo}CD4⁺ T cells in the spleen of chronic GVHD mice at 21 days after HCT produced IFN- γ , IL-13, IL-21 and IL-17 (see Figure 3D and page 9 bottom). We were not able to make similar measurements at later time points due to too few cells resulted from lymphopenia. We would also like to point out that intracellular cytokine staining requires in vitro stimulation with PMA plus Ionomycin of T cells for 4 hours, but this stimulation also changes PSGL-1 expression by CD4⁺ T cells. Therefore, in order to measure cytokine profile of PSGL-1^{lo}CD4⁺ T cells, we had to use flow cytometry sorting to purify the cell subset first, then stimulate them with PMA plus Ionomycin during in vitro culture and then measure their cytokine profile. The procedure requires large numbers of T cells.

3) Are the extra-follicular TFH-like cells promoting pathology via helping B cells make auto-Ab or are they making inflammatory cytokines that drive disease independent of Ab?

We observed that both PSGL-1^{lo}CD4⁺ T cells and tissue antibody deposition contribute to chronic GVHD pathogenesis, because blockade of PSGL-1^{lo}CD4⁺ T interaction via blockade ICOS/ICOS ligand interaction reduced serum anti-dsDNA concentration and ameliorated chronic GVHD tissue damage (Fig. 4 and 5). Reduction of PSGL-1^{lo}CD4⁺ T cell expansion by Stat3 deficiency in donor CD4⁺ T cells also decreased serum autoantibody production and tissue antibody deposition and ameliorated chronic GVHD tissue damage (Fig.S22 and newly added S23). These results indicate that PSGL-1^{lo}CD4⁺ T cells interact with B cells and augment B cell production of IgG antibodies, resulting in chronic GVHD tissue damage. Whether PSGL-1^{lo}CD4⁺ T cells alone can induce tissue damage or whether IgG antibody production from PSGL-1^{lo}CD4⁺ T interaction with B cells is required for chronic GVHD tissue damage will need to be addressed in the future studies.

4) The nomenclature used for the “no GVHD” groups are confusing in Figures 2, 6, 7. What are in the groups that say, for instance, “B-BCL-6^{+/+}-no GVHD”? What is the difference between the “B-BCL-6^{+/+}-no GVHD” and “B-BCL-6^{-/-}-no GVHD” groups?

To avoid confusion, we have clarified the labels in corresponding figure legends.

5) Figure 3C purports to show a difference in CXCR4 expression but both groups just have black boxes with no apparent differences.

We agree that the color difference in Figure 3C for CXCR4 was not clear. We have use real-time PCR to compare the difference and found it was statistically significant (see newly added Fig S12)

6) Figure 7A is missing a label

The label has been added.

7) When are spleens analyzed in Figure 7G?

60 days after HCT. This information has been added to the figure legend.

8) In figure 8, the cytokine arrow between pre-TFH and B cells should be going from the pre-TFH to B cells, not other way as shown.

This error has been fixed.

9) It should be noted that deletion of Bcl6 and Stat3 with CD4-cre affects Tregs, and specifically inhibit TFR cell generation.

We agree that deletion of BCL6 and Stat3 in CD4⁺ T cells decreases the number of follicular Treg cells (Chung et al: Nature medicine 17, 983-988 (2011); We et al: PloS one 11, e0155040 (2016). We observed that germinal centers were tiny or absent in recipients given TCD-BM only or TCD-BM + spleen cells from donors with specific deletion of Stat3 or BCL6 in CD4⁺ T cells. We also observed that neither lack of germinal center formation nor absence of follicular Treg cells has any significant impact on development of chronic GVHD development. These observations are consistent with the hypothesis that extrafollicular CD4⁺ T and B interaction plays a critical role in induction of chronic GVHD, and that peripheral Treg cells but not follicular Treg cells can regulate chronic GVHD development. This point has been added to our discussion (see page 19-20).

10) References should be given for the source of the Bcl6-flox mice and the Stat3-flox mice

References have been added (see page 22).

11) In discussion “T-B boarder” is written twice. It should be “T-B border”

This error has been corrected.

Reviewer #3 (ICOS, cytokine signaling, Tfh) (Remarks to the Author):

T follicular helper (Tfh) cells have emerged as a distinct subset/lineage of CD4⁺ T cells that are primarily

responsible for providing help to B cells during immune responses to TD Ag and mediating the differentiation of B cells into memory and plasma cells during GC reactions in secondary lymphoid tissues. The critical function of Tfh cells is evident from animal models and human diseases where Tfh cells are reduced/absent and humoral immunity is impaired, while on the other hand excessive numbers/production or function of Tfh cells has been associated with, and likely causes, autoimmune conditions such as SLE etc. however, there are other “types” of CD4 T cells that also provide help to B cells that are not strictly located within the B-cell follicles – these have been coined extrafollicular CD4⁺ T cells, and were first identified and characterized by Joe Craft in the setting of a murine model of autoimmunity. Since then, other groups have identified analogous cells in normal humoral immune responses, implicating these cells in the early stages of B-cell activation and differentiation (pre-germinal center).

In this current study, the authors have used numerous models of graft vs host disease and detailed physiological, developmental, cellular and functional analyses to reveal a critical role for extrafollicular CD4 T cells in disease pathogenesis. Importantly (but perhaps predictably) the generation of pathogenic extrafollicular CD4 T cells could be reduced by blocking ICOS/ICOS-L interactions, as well as genetic ablation of Bcl-6 or STAT3 in CD4 T cells. Overall, this study sheds substantial light on etiology of chronic GVHD, at least in murine models – it will need to be determined whether these findings are directly relevant to human GVHD. This notwithstanding, the study is well and comprehensively performed, and contains some important novel findings. Several comments follow that should be used as a guide to improve the novelty, mechanisms and conclusions of the findings.

1. The finding relating to the effects of STAT3 deficiency on disease need to be extended. First, what was the level of expression of Bcl-6 in Stat3-deficient CD4 T cells (and extrafollicular CD4 T cells specifically)? Second, Stat3 deficiency did not overcome acute GVHD, but greatly improved chronic GVHD. This raises questions about the nature and quality of extrafollicular CD4 T cells during these distinct phases of disease progression. Are these cells less likely to be pathogenic in the early/acute phase of GVHD? Third, Stat3 deficiency not only reduced pathogenic extrafollicular CD4 T cells but also increased Tregs. However, these latter cells were only assessed by phenotype – not function. Were the Stat3-deficient Tregs superior to WT Tregs at suppressing T cell activation and function? Was the increase in Tregs protective ie did disease severity worsen if Tregs were depleted from mice harbouring Stat3-deficient CD4 T cells?

We greatly appreciate the reviewer’s suggestion, and we have conducted new experiments to elucidate the role of Stat3 deficiency in CD4⁺ T cells in regulating chronic GVHD pathogenesis. We have also addressed this question by citing a previous publication.

Radojic et al (J. Immunol. 2010) showed that Stat-3 deficiency in donor CD4⁺ T cells effectively prevented induction of chronic GVHD in a model with B10D2 donors and MHC-matched BALB/c recipients. Prevention of chronic GVHD was associated with protection of the thymus and its ability to generate Treg cells, because depletion of Treg cells by anti-CD25 depletion abolished prevention of chronic GVHD in recipients given grafts from donors with specific Stat-3 deficiency in donor CD4⁺ T cells. Mechanisms that protect the thymus were not defined in that study.

In the current report, we showed that chronic GVHD in BALB/c recipients given B10D2 transplants lost lymphofollicles and germinal center formation (Fig. S7), as observed with with BALB/c recipients given MHC-mismatched C57BL/6 transplants. We found that specific Stat3 deficiency in donor CD4⁺ T cells protected the thymus and increased the numbers of Foxp3⁺ Treg cells in the periphery in BALB/c recipients given C57BL/6 transplants (Fig. S19 and Fig. S24). These findings are consistent with the results reported by Radojic et al.

Furthermore, we found that thymus protection in BALB/c recipients given C57BL/6 donor spleen cells with specific Stat3 deficiency in CD4⁺ T cells (CD4-Stat3^{-/-}) was associated with a significant reduction of expansion

of extrafollicular PSGL-1^{lo}CD4⁺ T cells in the spleen and a significant reduction of PSGL-1^{lo}CD4⁺ T cell infiltration in the thymus early after HCT (newly added Fig. 8). Reduced infiltration of PSGL-1^{lo}CD4⁺ T cells allowed full recovery of thymus in recipients given CD4-Stat3 deficient transplants (newly added Fig. 8). Donor CD4⁺ T cells mediate thymus damage early after HCT via FasL (Na et al JCI 2010). Add-back of PSGL-1^{lo}CD4⁺ T cells worsened thymic damage in recipients given CD4-BCL6^{-/-} transplants that cannot give rise to PSGL-1^{lo}CD4⁺ T cells (newly added Fig. S17). This information has been added to result and discuss sections (see pages 14-15 and 19).

2. Several of the Figures contain results from data for gene expression from RNA Seq. these data need to be confirmed by Western blot or FACS. Eg Fig 3C – for CXCR4, CXCR5 and CCR7; Fig 4A – for many of the surface receptors listed; Fig 7 for STAT3 and BCL6. Also for Fig 7A, it is not indicated which panels correspond to which experimental mice.

For RNA-seq data, each sample represent the sorted PSGL-1^{lo}CD4⁺ T cells pulled from 8 mice, and two replicate experiments are shown. The related RNA-seq data are validated with real time PCR: newly added Fig. S12 for Fig. 3C assessment of CXCR4, CXCR5 and CCR7; newly added Fig. S15 for Fig. 4A assessment of ICOS, PD-1, PD-L1 and CD80; newly added Fig. S18 for Fig. 7A assessment of Stat3 and BCL6.

3. For the expts for Fig 5D, it is possible that the reduced level of detected ICOS expressed by the labelled cells by FACS in the mice receiving the anti-ICOS mAb reflects masking by the blocking ICOS mAb that was still bound to the cells. This needs to be determined. Also for this data the treatment regime was extreme – blocking ICOS Ab injected every 2nd day for 45 days. Was this necessary? Was there a shorter time frame in which the ICOS/ICOS-L interaction could be blocked and a physiological effect still be detected?

Yes, the anti-ICOS mAb staining in Figure 5D reflects masking by the blocking ICOS mAb that was still bound to the cells, as demonstrated by the binding of anti-rat IgG2b staining in Figure 5C.

We agree with the reviewer that the dosing of anti-ICOS treatment appeared to be extreme. Several groups have shown that anti-ICOS mAb ameliorates acute and chronic GVHD (Mollweide A et al: Klin Padiatr. 2009; 221:344. Fujimura et al: Pediatr Transplant. 2010; 14: 540. Flynn R et al: Blood 123, 3988-3998, 2014). The major goal of our experiments is to test whether ICOS/ICOS ligand interaction plays an important role in regulating expansion of extrafollicular PSGL1^{lo}CD4⁺ T cells, and we adopted the extreme dosing from a previous report (Flynn R. et al: Blood 2014), because we wanted to see the strongest possible effect. The results clearly show that blockade of ICOS/ICOS ligand interactions reduces the expansion of extrafollicular PSGL1^{lo}CD4⁺ T cells. Further exploration of the best dosing and duration of anti-ICOS treatment was not the focus of current studies, and we did not carry out experiments to address this question. The above points have been incorporated to discussion (see page 17, last paragraph).

REVIEWERS' COMMENTS:

Reviewer #1 (Remarks to the Author):

The authors have clarified some issues.

Thank you!

Reviewer #2 (Remarks to the Author):

As I wrote previously, the major claims of the paper are showing that extra-follicular CD4 T cells are involved in cGVHD pathogenesis and that cGVHD can be blocked by inhibiting these cells via an ICOS blockade, or via genetic disruption of the transcription factors Bcl6 or Stat3 in T cells. These findings are important for the understanding of cGVHD and also help to clarify respective roles of germinal center-associated TFH cells and extra-follicular TFH-like cells.

The revised manuscript addresses my most pressing concerns, but I still find it disconcerting that the authors do not have flow cytometry comparing Tfh versus the supposed extra-follicular Tfh-like cells. However, the RNA expression data does address CXCR5 expression and basically addresses the issue.

The legends are better, but in Figure 1, it should be laid out clearly what Group 1 is versus Group 2 versus Group 3. Also, TCD should be defined somewhere in the main manuscript.

In summary, the main conclusions of the paper are significant for the field, and convincingly presented. Statistics seem fine and there is probably enough detail given for reproduction by other researchers.

Thanks for the suggestion. We have defined Group 1, group 2 and Group 3 in Figure 1 legend and all figures with similar situation. We have also defined TCD (T cell-depleted) in the text at its first appearance.

Reviewer #3 (Remarks to the Author):

The authors have adequately addressed most of the concerns raised in the original review of their manuscript. However, they did not assess/confirm differential gene expression by FACS, as requested - rather they have confirmed these differences by qPCR. This should suffice - tho it would have been preferable to see either protein expression (ie FACS analyses) or simply for the authors to state that they felt the data presented as qPCR was suitable.

Thanks for the suggestion. We added the statement "as measured by RNA seq analysis and suitably confirmed by real-time PCR" to the corresponding text.